

# Enhancing Climate Model Performance through Improving Volcanic Aerosol Representation

Ziming Ke[1], Qi Tang[1], Jean-Christophe Golaz[1], Xiaohong Liu[2], and Hailong Wang[3]

[1]Lawrence Livermore National Laboratory, Livermore, CA, USA
[2]Texas A&M University, College Station, TX, USA
[3]Pacific Northwest National Laboratory, Richland, WA, USA

Correspondence to: Ziming Ke (ke2@llnl.gov)

**Abstract**

An accurate representation of Earth's surface temperature is crucial for simulating climate change. Yet many climate models struggle to reproduce the evolution of historical temperature records, especially after the major 1963 Mt. Agung volcano eruption. This study investigates whether the method of specifying the volcanic forcing could be contributing to this bias using the Energy Exascale Earth System Model (E3SM). The CMIP6 protocol represents volcanic eruptions through simplified radiative forcing, neglecting the interaction between volcanic aerosols and clouds. Here we adopt a new approach based on an updated volcanic eruption inventory, which includes volcanic sulfur dioxide emissions and hence allows for a more realistic representation of subsequent physical processes that involve volcanic aerosols. With this new approach, E3SM simulates slightly warmer surface temperatures and improved interannual variability during years 1940-1980 compared to the standard CMIP6 approach. The improvements mainly stem from two factors: 1) the inclusion of volcanic aerosol-cloud interactions, which reduces aerosol indirect effect by volcanic quiescent warming effect, and 2) the more accurate representation of volcanic eruptions after 1963, which leads to less volcanic aerosol cooling. Overall, this study highlights the importance of more accurate volcanic forcing in improving climate simulation and is strongly in favor of an emission-based volcanic forcing treatment.



**1. Introduction**
Volcanic eruptions, as manifestations of natural radiative forcing, play a crucial role in modulating climate changes
(e.g. Chim et al., 2023; Hegerl et al., 2003). Numerous studies have demonstrated their significant impacts on
Earth's climate. For example, the eruption of Tambora (Indonesia) in April 1815 led to the 'Year Without a
Summer' of 1816 in Europe and North America — which extended to several years in China—as well as severe
disruptions to the Indian monsoon and to other global climate patterns (Raible et al., 2016). The 1991 eruption of
Mt. Pinatubo resulted in a peak top-of-the-atmosphere radiative forcing of roughly 3-4 W/m2 and cooled global
temperatures up to 0.4 °C (e.g. Dhomse et al., 2014; Mills et al., 2017; Ramachandran et al., 2000; Rieger et al.,
41 2020)
Intensive volcanic eruptions emit a variety of gases and particles into the stratosphere. The emitted sulfur dioxide
($SO_2$) forms sulfate aerosols through atmospheric chemical reactions, which are the primary drivers of climate
perturbation (i.e. Dhomse et al., 2014; Mills et al., 2016). Water vapor is scarce in the stratosphere. Sulfate aerosols
can persist for months to years due to lack of wet removal as compared to days in the troposphere (Mills et al.,
2017). By scattering incoming solar radiation, these sulfate aerosols induce cooling at the Earth surface while
simultaneously absorbing longwave radiation, thereby warming the surrounding air (Schmidt et al., 2018). This
effect caused specifically by volcanic sulfate aerosol is volcanic aerosol-radiation interactions (VARIs).
Additionally, akin to anthropogenic sulfate aerosols, volcanic sulfate particles can act as cloud condensation nuclei
(CCN), facilitating the formation of cloud droplets and changing of cloud albedo properties (Schmidt et al., 2012).
This is volcanic aerosol-cloud interactions (VACIs).
In the CMIP6 simulations, many climate models underestimated global mean surface temperature in the middle of
the 20th century mainly due to cloud enhancement caused by aerosol-cloud interactions (Flynn and Mauritsen, 2020;
Zhang et al., 2021). Notably, in the case of E3SM version 2 (E3SMv2), the simulated historical surface temperatures
exhibit a distinct low bias since year 1940 (Golaz et al., 2022). This temperature bias becomes more pronounced
after the eruption of Mt. Agung in 1963. This temporal alignment of temperature low bias with volcanic eruption
events motivates our investigation into whether the model's representation of volcanic activity has been contributing
to the temperature low bias.
Importantly, the impact of volcanic eruptions can extend beyond isolated events (Chylek et al., 2020; Cole-Dai,
2010; Robock, 2000). Schmidt et al. (2012) emphasized the importance of volcanic aerosols induced aerosol-cloud
interactions in the pre-industrial (PI) and present-day (PD) baseline simulations. When factoring in the indirect
effect of volcanic sulfate aerosols in PI and PD simulations, the historical aerosol's indirect radiative effect was
diminished. It is worth noting that most of climate models participating in CMIP6 didn't represent the VACIs in
their aerosol-cloud parameterizations. Adding the same amount of volcanic $SO_2$ in PI and PD simulations, the
relative Cloud Droplet Number Concentration (CDNC) changes, $(PD_{cdnc}-PI_{cdnc})/PI_{cdnc}$, become less due to higher PI
background aerosol concentration. Furthermore, as volcanic emissions fluctuate over time, opposite to relatively
stable anthropogenic emissions, their impact on aerosol-cloud interactions also varies with time. During volcanic
quiescent periods, with eruptions below the historical average, the reduced volcanic emissions could partially offset
anthropogenic $SO_2$ emission increases, resulting in a relative warming effect. Conversely, during volcanic active
periods, additional volcanic emissions could augment total sulfate aerosol burden on the top of anthropogenic
emissions. These **volcanic quiescent warming effect** and **volcanic surplus cooling effect** underscore the
importance of considering volcanic aerosols in climate simulations, which will be described in detail in Section 2
and discussed in Section 3.
Recognizing the limitations of the CMIP6 volcanic forcing treatment, here we propose a new methodology, which
involves using volcanic $SO_2$ emissions to replace prescribed volcanic stratospheric forcing, thereby capturing both
the VARIs and VACIs effects. By incorporating the averaged volcanic $SO_2$ emissions in the PI control simulations,
the volcanic quiescent warming and surplus cooling effects can be appropriately represented in subsequent historical
simulations. These model developments have been integrated into the version 3 of E3SM (see Figure 1, right panel).





To assess whether the new volcanic treatment improves E3SMv2 simulated climate, we conducted new historical
simulations by implementing the updated treatment in E3SMv2, which includes new PI control simulations and
transient simulations spanning from 1850 to 2014. Further details regarding these simulations are outlined in the
methods and experimental sections (Section 2). Result analysis is presented in Section 3, followed by conclusions in
Section 4.

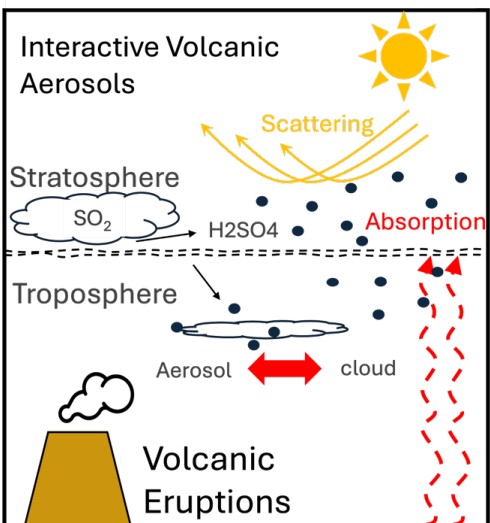

Figure 1. Volcanic forcing representations in E3SM: Prescribed stratospheric scattering and absorption following
CMIP6 protocol (left) and the interactive volcanic aerosols used in this study (right).

## 2. Methods and Experiments
### 2.1 The Volcanic Forcing Representation in E3SMv2

E3SMv2 is the state-of-the-art earth system model including a atmosphere model at 110 km horizontal resolution, a
land model at 165 km horizontal resolution, a 0.5°- horizontal resolution river routing model, and an ocean and sea
ice model with mesh spacing ranging from 60 km in mid-latitudes to 30 km at the equator and poles. The
atmosphere component, E3SM Atmosphere Model (EAM) v2, comprises 72 vertical layers extending to
approximately 60 km. Within EAMv2, the Cloud Layers Unified By Binormals (CLUBB) parameterization (Guo et
al., 2015) handles the subgrid turbulent transport and the macrophysics of stratiform and shallow cumulus clouds,
while the planetary boundary layer (PBL) depth is diagnosed following the scheme by Holtslag and Boville
(Holtslag and Boville, 1993). Deep convection is represented by a scheme developed by (Zhang and McFarlane,
1995), with an improved trigger function combining the dynamic Convective Available Potential Energy (dCAPE)
trigger (Wang et al., 2020b) and unrestricted air parcel launch level (ULL). Grid-scale cloud microphysical
processes are parameterized using the version 2 of the Morrison and Gettelman (Morrison and Gettelman, 2008)
microphysics scheme. E3SMv2 demonstrates enhanced performance compared to E3SMv1, with nearly double the
computational speed and improvements in various metrics such as precipitation and cloud representation. Notably,
its climate sensitivity is substantially lower, with an equilibrium climate sensitivity of 4.0 K, as opposed to the less
plausible value of 5.3 K in E3SMv1. However, similar to many other CMIP6 models E3SMv2 simulates a low
surface temperature bias in the middle of 20$^{th}$ century, primarily due to excessive aerosol radiative forcing (Golaz et
al., 2022).



Following the CMIP6 protocol, E3SMv2 employs prescribed volcanic shortwave extinction and longwave
absorption above the tropopause (Golaz et al., 2022) (Zanchettin et al., 2016). Particularly, the stratospheric aerosol
extinction and absorption are overwritten by prescribed values at each time step. For the period spanning 1979-2014,
data predominantly rely on assimilated satellite data from sources like the Stratospheric Aerosol and Gas
Experiment (SAGE), SAGEII, the Stratospheric Aerosol Measurement (SAM), the Cloud-Aerosol Lidar and
Infrared Pathfinder Satellite Observation (CALIPSO), and the Optical Spectrograph and InfraRed Imager System
(OSIRIS), with the Cryogenic Limb Array Etalon Spectrometer (CLAES) data utilized for gap-filling in cases of
missing data (Rieger et al., 2020; Thomason et al., 2018). During the period from 1850 to 1978, particularly during
volcanically quiescent periods, the monthly mean background aerosol data measured by SAGE II (during the
volcanic quiescent period of 1996-2005) is utilized. The volcanic eruption contribution is then calculated using the
two-dimensional sulfate aerosol model developed at the Atmospheric and Environmental Research Inc., Lexington,
MA, USA (AER-2-D). The AER-2-D model has sulfuric acid aerosol microphysics in a global domain with 9.5°
horizontal resolution and 1.2 km vertical resolution. The aerosol microphysics scheme has 40 size bins spanning the
range 0.4 nm to 3.2 $\mu$m. There is no interaction between aerosols, radiation forcings, and dynamics and the
dynamical fields, such as U, V, and T, for all simulated cases are based on Pinatubo eruption climatology (1991).
Additionally, stratospheric AOD is calibrated using the photometer data whenever available; otherwise, the best
estimate of sulfur ejection is utilized for the volcanic contribution, often estimated from proxies such as ice core data
(Arfeuille et al., 2014). For the PI control simulation, the volcanic quiescent background values are used.
This study focuses on volcanic activities during the year 1940-1979 period (the reason will be described in section
2.2). During this period, Arfeuille et al. (2014) recorded two volcanic eruptions (Table 1). For the Agung (1963)
eruption, AER-2-D model evenly injected $SO_2$ in the 15°S-0° and 0°-15°N regions of Southern and Northern
Hemispheres, respectively. For the Fuego (1974) eruption, $SO_2$ was injected evenly in the 0°-15°N band only
(Arfeuille et al., 2014). Compared to injecting emissions at limited grids, evenly distributing the emission in a broad
latitude band dilutes the $SO_2$ concentration and consequently results in smaller particle sizes and thus higher
efficiency of scattering the solar radiation and prolonged aerosol lifetime (Niemeier et al., 2019; Timmreck et al.,
144  2010).
**2.2  The interactive volcanic aerosol treatment**
In E3SMv2, the aerosol process is represented by 4-mode version of the modal aerosol model (MAM4) (Liu et al.,
2012; Wang et al., 2019), which is a comprehensive approach to simulate aerosol particles in the Earth system. It
encompasses four distinct aerosol modes representing different aerosol types and sizes: Aitken mode, accumulation
mode, coarse mode, and primary carbon mode for black carbon and primary organic carbon particles emitted
directly into the atmosphere. This model accounts for aerosol processes such as emissions, transport, chemical
transformation, and removal.
In the current version of MAM4, there are six aerosol species represented: sulfate, black carbon, organic carbon,
dust, sea salt, and secondary organic aerosols. In E3SMv2, sulfate aerosols primarily originate from the
condensation of $H_2SO_4$ gas as well as aqueous phase production in cloud water. The model utilizes a simple gas-
phase chemistry package to calculate the formation of $H_2SO_4$, incorporating prescribed oxidant, hydroxyl radical
(OH), to oxidize $SO_2$ and DMS gases in the atmosphere.
It is worth noting that the MAM4 in E3SMv2 hasn't been designed to accurately reproduce the volcanic aerosol
direct effect caused by volcanic eruptions. Modifications are needed to well reproduce the Mt. Pinatubo's (1991)
aerosol direct impact on shortwave forcing, compared to observations (Mills et al., 2014). But such modifications
caused unexpected drawbacks of ice cloud formations over upper troposphere and lower stratosphere (Visioni et al.,
2017). The remedy efforts for both CESM and E3SM will be represented in a following-up paper that documents a
new development of adding a stratospheric sulfate mode on top of MAM4 (Ke et al., in preparation). Furthermore, it
is important to use unchanged MAM4 and E3SMv2 configurations to provide an apple-to-apple comparison to
evaluate the impacts of the change of volcanic aerosol representation on simulated aerosol direct and indirect effects
during middle of 20[th] century.
To introduce interactive volcanic aerosols into E3SMv2, we utilize the volcanic Emissions from Earth System
Models (volcanEESM) dataset, which serves as a source of volcanic $SO_2$ emissions (Danabasoglu et al., 2020; Neely
and Schmidt, 2016). This dataset, funded by the NCAR/UCAR Atmospheric Chemistry and Modeling Visiting




Scientist Program and the University of Leeds School of Earth and Environment, provides detailed information on
historical volcanic eruptions, including dates, locations, injection height ranges, and $SO_2$ emission amounts. Given
that E3SMv2 lacks comprehensive stratospheric chemistry for processing $SO_2$ gas, we employ the simplified
chemistry package where volcanic $SO_2$ is oxidized using prescribed OH concentrations derived from the historical
monthly mean from the CESM-WACCM simulations. Past research has demonstrated that this approach yields
reasonable results with high efficiency compared to models employing the comprehensive stratospheric chemistry
(Smith et al., 2014). We validate this approach by comparing the simulated interactive stratospheric aerosol optical
depth (SAOD) in E3SMv2 with SAOD produced using the default method (see Figure 2).
Figure 2 depicts the simulated SAOD based on volcanEESM dataset and CMIP6 default method, respectively.
Generally, the two simulated SAOD curves align closely in terms of eruption timing and intensity. However, notable
discrepancies emerge between 1940 and 1980 (black dashed box). Specifically, volcanEESM records two moderate-
intensity eruptions during 1940 to 1960, whereas no eruptions are recorded in the CMIP6 volcanic dataset for this
period. Additionally, the CMIP6 shows higher SAOD values than those predicted from the volcanEESM for the Mt.
Agung (1963) eruption and the two subsequent eruptions, which were not recorded in the CMIP6 document (Table
1, Arfeuille et al., 2014). These significant disparities motivate our study to investigate the impact of the
volcanEESM inventory on simulated climate compared to that using the default E3SMv2 model with the CMIP6
volcanic dataset. The volcanic eruptions during 1940 to 1980 from CMIP6 and volcanEESM are presented in Tables
1 and 2, respectively.

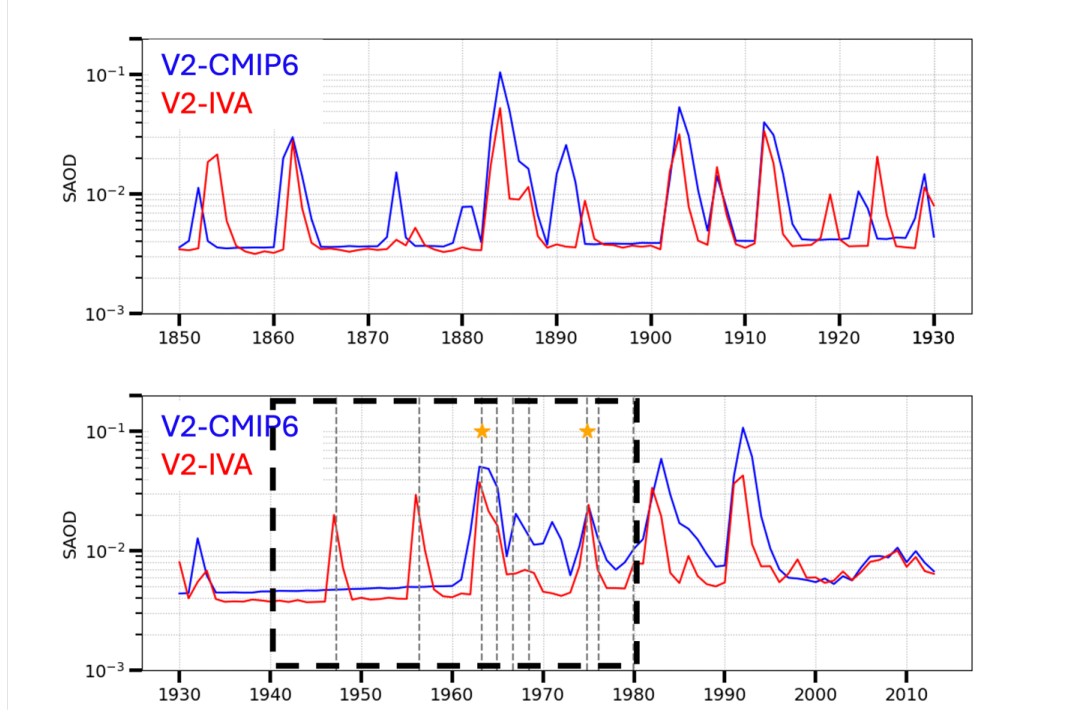

Figure 2. The simulated stratospheric AOD (SAOD) by E3SMv2 using different volcanic representations. The
E3SMv2 with CMIP6 prescribed volcanic scattering (V2-CMIP6) is shown in blue line, while the E3SMv2 with
interactive volcanic aerosol treatment (V2-IVA) is shown in red line. During the year 1940-1980 period, the
volcanic eruptions recorded by CMIP6 data are marked by orange stars, while the eruptions recorded by
volcanEESM are marked by grey dashed lines.



Table 1. Recorded eruptions based on Arfeuille2014 (CMIP6).

|  | Injection Height (km) | NH (SO2 Tg) | SH (SO2 Tg) |
|---|---|---|---|
| 1963 Agung | 27 | 3.4 | 6.5 |
| 1974 Fuego | 33.5 (as Pinatubo) | 2.3 | 0.0 |






Table 2. Recorded eruptions from volcanEESM during year 1940-1979.

| YYYY/MM/DD | LAT | LON | ALTMIN | ALTMAX | SO2(TG) |
|---|---|---|---|---|---|
| 1947/03/29 | 64.0 | 339.3 | 15.0 | 19.6 | 2.3 |
| 1956/03/30 | 56.0 | 160.6 | 15.5 | 18.5 | 3.9 |
| 1963/03/17 | -8.3 | 115.5 | 18.0 | 20.0 | 7.5 |
| 1964/11/12 | 56.7 | 161.4 | 15.0 | 19.6 | 2.3 |
| 1966/08/12 | 3.7 | 125.5 | 15.0 | 19.6 | 0.8 |
| 1968/06/11 | -0.4 | 267.5 | 15.0 | 19.6 | 0.8 |
| 1974/10/10 | 14.5 | 268.1 | 16.7 | 21.3 | 3.0 |
| 1976/01/22 | 59.4 | 205.6 | 7.0 | 10.0 | 0.8 |
| 1979/11/13 | -0.8 | 268.8 | 1.5 | 14.0 | 1.2 |

**2.3 Mechanisms of volcanic aerosols to affect climate**
Studies have highlighted that among the various gases and ash particles emitted during volcanic eruptions, the
primary climate impact stems from sulfate aerosols formed through atmospheric chemical reactions from emitted
$SO_2$ gas. Depending on the eruption's intensity, $SO_2$ can be injected into either the troposphere or the stratosphere. In
the troposphere, where moisture is abundant, sulfate aerosols are swiftly removed through the wet scavenging
process, with a lifespan of 2 to 5 days. However, if emissions reach the stratosphere, where water vapor is scarce,



sulfate aerosols are primarily removed through gravitational settling and dry deposition, prolonging their lifespan for months or even years (Cole-Dai, 2010; Robock, 2000).

The literature has extensively documented the direct and indirect effects of sulfate aerosols on climate (Bauer and Menon, 2012; Boucher et al., 2013; Ghan et al., 2012; Grandey et al., 2018; Liu et al., 2012; Penner et al., 2001; Wang et al., 2020a; Zhang et al., 2022). Directly, sulfate aerosols scatter the incoming solar radiation, cooling the atmosphere below, while they simultaneously absorb the longwave radiation, warming the surrounding air. Volcanic sulfate aerosols reflect solar radiation in the stratosphere, reducing net shortwave forcing both at the top of the atmosphere and at the surface, while also absorbing longwave radiation from below, warming the stratosphere. Hereafter this effect is referred to as the VARIs. Furthermore, sulfate particles descending from the stratosphere to the lower troposphere act as cloud condensation nuclei (CCN), facilitating the cloud formation. Increased CCN concentration results in smaller and more numerous cloud droplets, making clouds brighter and more reflective. Hereafter this effect is referred to as the VACIs.

The aerosol radiative forcing in the historical period primarily arises from anthropogenic $SO_2$ emissions, which increases aerosol concentrations, initiating aerosol-cloud interactions (ACIs) that amplify cloud formation and prolong cloud lifetimes (Chylek et al., 2020; Zhang et al., 2021, 2022). Consequently, clouds scatter more incoming shortwaves (see Figure 3, upper panel). Simulations including averaged volcanic $SO_2$ emissions in preindustrial control scenarios showed a rise in background aerosol concentrations at PI. Consequently, the same increase in anthropogenic $SO_2$ emissions induces weaker ACIs, resulting in relatively less cloud droplet increase and reduced shortwave scattering by clouds. Ultimately, this leads to a warmer surface temperature compared to the former scenario (see Figure 3, bottom panel).

However, the historical periods, when volcanic emission amount equal to the historical average are rare. It's more common to observe volcanic quiescent and active periods. During volcanic quiescent periods, during which volcanic emission amount below the historical average, the volcanic aerosol change is negative compared to the baseline. This negative change can partially offset anthropogenic emission growth, resulting in reduced historical aerosol change compared to the control case, where volcanic aerosols are not considered. Consequently, a less amplified cloud cooling effect is expected during volcanic quiescent periods compared to the control, which we term as the **volcanic quiescent warming effect**.

Conversely, during volcanic active periods, when volcanic aerosol emissions exceed the historical mean, the volcanic aerosol change is positive. In this scenario, an enhanced aerosol indirect effect is expected, leading to increased cloud cooling compared to the control case. This effect is termed as the **volcanic surplus cooling effect** in this study.





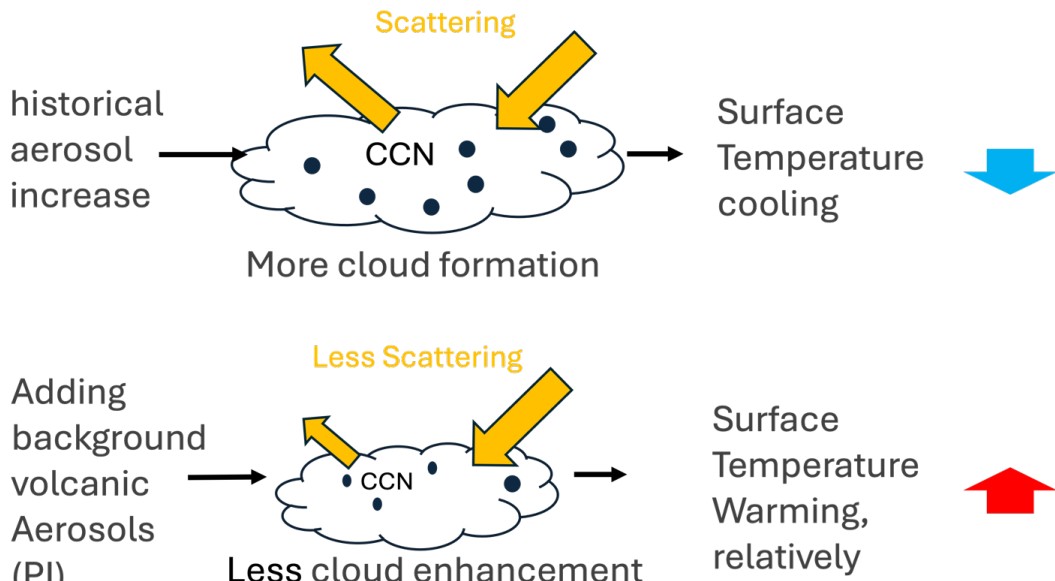

Figure 3. Schematic illustrating the aerosol-cloud interaction mechanism driving historical aerosol forcing. Upper panel: Aerosol-cloud interactions (ACIs) amplify cloud formation and prolong cloud lifetimes, increasing shortwave scattering. Bottom panel: Incorporating volcanic $SO_2$ emissions shows a rise in background aerosol concentration, relatively diminishing cloud formation and reducing shortwave scattering, resulting in a relatively warmer surface temperature.

## 2.4 The Experimental Design

### 2.4.1 Averaged volcanic emission in pre-industrial control simulation

The CMIP6 protocol recommends using averaged volcanic forcing in the historical period in the PI control simulations. The average volcanic $SO_2$ emission is $2.26 \times 10^{-8}$ Tg s$^{-1}$, equivalent to 0.7 Tg year$^{-1}$, calculated by averaging emissions from all eruptions between 1850 and 2014. To determine the horizontal emission distribution, we assume a normal distribution along latitude and even distribution along longitude. Using each eruption amount as a weight, the weighted mean emission latitude is 20.67º north, with a standard deviation of 28.83º. Vertically, the mean injection height has an upper limit of 18 km and a lower limit of 14 km.

### 2.4.2 Control ensembles and V2-IVA and V2-IVA-NPI ensembles

The V2-CMIP6 (control) run comprises a 5-member ensemble of E3SMv2 coupled historical transient simulations spanning from 1850 to 2014, conducted and archived by Golaz et al., 2022. In contrast, the V2-IVA experiment investigates the influence of interactive volcanic aerosols on historical transient simulations by replacing the default prescribed volcanic stratospheric forcing with volcanic $SO_2$ emissions in E3SMv2. This experiment underwent a 100-year spin-up under the same preindustrial (PI) control configuration as V2-CMIP6, except with interactive volcanic treatment, utilizing averaged volcanic emissions from 1850 to 2014 (see section 2.4.1). Following the model spin-up, one member simulation is conducted from 1850 to 2014, with additional two members conducted from 1940 to 2014 to minimize noise in coupled simulations. By comparing V2-IVA to V2-CMIP6, the impact of volcanic treatments on simulated climate can be assessed.





Additionally, to evaluate the influence of background volcanic aerosols on historical transient simulations, the V2-
IVA-NPI experiment was conducted, which is identical to V2-IVA but the averaged volcanic emissions removed
during its 100-year PI control spin-up. Like V2-IVA, V2-IVA-NPI also has one member conducted from 1850 to
2014, with additional two members conducted from 1940 to 2014 to minimize noise in coupled simulations. These
three experiments are summarized in Table 3.
**Table 3**. Experiment configurations.

|  | Simulation Type | Historical Volcanic Forcing | piControl Volcanic Setting |
|---|---|---|---|
| V2-CMIP6 (E3SMv2 default) | V2 archived 5 members: 1850-2014 | Prescribed in Stratosphere (following CMIP6) | Prescribed in Stratosphere (volcanic quiescent background) |
| V2-IVA | 1 member 1850-1940 3 members 1940-2014 | Interactive treatment (using VolcanEESM) | Averaged emission (1850-2014) |
| V2-IVA-NPI | 1 member 1850-1940 3 members 1940-2014 | Interactive treatment (using VolcanEESM) | No volcanic emission |





### 3. Results
#### 3.1 Simulated Sulfate Aerosols and Forcing Fields

VolcanEESM recorded eight eruptions during the years spanning 1940-1979 (see Table 2). These eruptions are directly reflected in sulfate aerosol concentrations simulated by the V2-IVA experiment (see Figure 4, panel a). Prior to these eruptions, the background sulfate aerosol concentration between 100 and 50 hPa was approximately 0.1 $\mu$g/kg. Helka (1947) and Bezymianny (1956), emitting 2.3 and 3.9 Tg $SO_2$ gas respectively, induced spikes in sulfate aerosol concentrations, with global mean concentration peaks reaching up to 7 and 12 $\mu$g/kg in the stratosphere, respectively. The eruption of Mt. Agung in 1963 with a $SO_2$ emission of 7.5 Tg, caused a peak global mean concentration of up to 20 $\mu$g/kg between 100 and 10 hPa. Subsequent to the Mt. Agung (1963) eruption, three eruptions resulted in high aerosol concentrations lingering in the stratosphere until 1972, with eruptions in 1974 and 1976 sustaining global mean concentrations above 0.5 $\mu$g/kg for additional four years.

In addition to their significant amounts in the stratosphere, volcanic sulfate aerosols gradually descended into the troposphere. As V2-CMIP6 did not account for volcanic aerosols, the sulfate aerosol difference between V2-IVA and V2-CMIP6 illustrates the descent of these aerosols into the troposphere, which is depicted in Figures 4b to 4e. Four out of the eight eruptions recorded by volcanEESM (1947, 1956, 1964, 1976) occurred in northern hemisphere high latitudes (above 50° N), while the other four occurred in the tropical regions (20° S to 20° N, see Table 2). Strong sulfate aerosol footprints were observed in the troposphere (below the tropopause, gray lines) in northern high latitudes (Figure 4c) compared to tropics (Figure 4d) and southern high latitudes (Figure 4e). Despite no eruptions occurring in southern hemisphere high latitudes, volcanic aerosols tended to descend more over these regions compared to tropical regions due to the Brewer-Dobson circulation. Overall, a substantial amount of sulfate aerosols reached the troposphere from the stratosphere, highlighting the potential aerosol-cloud interactions.

SAOD describes the impact of aerosols on the optical properties of the atmosphere in the stratosphere. The simulated SAOD from the V2-IVA ensemble is shown in Figure 5, upper panel. Prior to eruptions, the background SAOD values were approximately 0.008 over high latitudes and 0.002 over the tropics. The volcanic eruptions of Helka (1947) and Bezymianny (1956) elevated SAOD to 0.06 and 0.13 over northern hemisphere high latitudes (compared to Figure 5 in Danabasoglu et al., 2020). Since these two volcanic eruptions were absent in V2-CMIP6, the two red spikes emerged when comparing V2-IVA with V2-CMIP6 (Figure 5 lower panel). Despite their relatively small magnitudes, the impact of these two volcanoes was limited to two years and north of 30 degrees in the Northern Hemisphere.

For the Mt. Agung eruption in 1963, V2-IVA SAOD displayed a clear spike spreading from the tropics to the South Pole, with peak values exceeding 0.2, consistent with previous studies (Dhomse et al., 2020; Niemeier et al., 2019) . Due to a lower strength recorded by volcanEESM, the simulated SAOD in V2-IVA was approximately 0.03 lower than in V2-CMIP6 (Figure 5, lower panel). Additionally, V2-CMIP6 simulation indicated three events with slightly higher SAOD than V2-IVA in 1967, 1972, and 1974, spanning from the tropics to southern hemisphere high latitudes, while V2-IVA recorded an extra eruption in 1976 in northern hemisphere high latitudes. Consequently, V2-IVA simulated two moderate volcanic eruptions during the 1940-1959 period and a relatively dimmer volcanic impact over the 1960-1979 period.

Aerosol extinction vertical profiles measure the scattering and absorption of solar radiation by aerosols. Figure 6 examines the difference in simulated extinction between V2-IVA and V2-CMIP6 across time and pressure levels. In the V2-IVA simulation, extinction resulted from simulated aerosol scattering and absorption effects, including volcanic aerosols, whereas in V2-CMIP6, the extinction caused by volcanic eruptions was prescribed. In the panel for global mean, distinct red stripes caused by the Helka (1947) and Bezymianny (1956) eruptions extend from the stratosphere into the upper troposphere, coinciding with the comparison of sulfate aerosol concentrations (Figure 4). This is attributed to V2-IVA's interactive treatment of volcanic aerosols, allowing for their light extinction effect to penetrate below the stratosphere as particles descend into the troposphere, a more realistic representation compared to the prescribed treatment in V2-CMIP6. The light extinction of the Helka (1947) and Bezymianny (1956) eruptions is primarily observed between 50 and 350 hPa globally. In northern high latitudes, the impact of the Bezymianny (1956) eruption could extend to the middle to lower troposphere below 500 hPa, whereas its impact over the tropics was relatively weaker.



Regarding the Mt. Aqung eruption, V2-IVA simulated a weaker response above 100 hPa compared to V2-CMIP6,
resulting in negative values. However, V2-IVA simulated stronger extinction between 100 and 300 hPa compared to
V2-CMIP6, as the injection was concentrated in the middle to lower stratosphere (18-20 km) in V2-IVA (see Table
2). For the eruptions subsequent to Mt. Aqung, V2-IVA simulated dimmer eruptions compared to V2-CMIP6 on a
global average. In detail, V2-IVA simulated slightly stronger extinctions over northern high latitudes, while showing
dimmer scattering over the tropics and the southern hemisphere.
Figure 7 shows the shortwave and longwave radiative forcings from V2-IVA and V2-CMIP6 ensembles at the top of
the model. In panel *a*, the shortwave forcing under clear-sky conditions is shown. As anticipated, the Helka (1947)
and Bezymianny (1956) eruptions caused clear-sky cooling effects in the V2-IVA simulations, with global mean
radiative forcing dropping by 0.6 and 1.0 W/m², respectively. These drops took more than a year for the forcing to
recover, whereas there were no such drops in the V2-CMIP6 simulations. For the Mt. Aqung (1963) eruption, V2-
IVA simulated a drop of 2.3 W/m², while V2-CMIP6 simulated a drop of 2.7 W/m². Three years later (1966), both
simulations showed the same level of forcing, with a small discrepancy appearing again during 1967-1974 due to
differences in volcanic forcing mentioned in Figures 5 and 6. This reduced volcanic effective ARI forcing agrees
with previous studies which pointed out that CMIP6 volcanic aerosols' direct radiative forcing (VARI) would be too
strong (Chylek et al., 2020; Dhomse et al., 2020; Niemeier et al., 2019).
Panel *c* examines the shortwave forcing under all-sky conditions. The discrepancy caused by the Helka (1947)
eruption becomes less clear. This indicates that the volcanic scattering effect over high latitudes partially is offset by
cloud warming effect. The dimmer eruptions simulated by V2-IVA compared to V2-CMIP6 in the 1960s are
consistent with panel *a*.
In panel *d*, we present the outgoing longwave forcing at the top of the model. There is no clear signal for the Helka
(1947) eruption, and the weaker values caused by Bezymianny (1956) simulated by V2-IVA indicate longwave
warming, partially offsetting the cooling in the shortwave spectrum (panel *c*). After 1963, V2-IVA simulated more
outgoing longwave forcing compared to V2-CMIP6.
The above analysis looked at how volcanic eruptions and sulfate aerosols affect radiative forcings, comparing
between V2-IVA and V2-CMIP6. It examined sulfate concentration, SAOD, extinction profiles, and radiative
forcings at the top of the model to highlight how the change of the volcanic representation lead to variations in
radiative forcings and aerosol concentration fields. In the next section, we will examine the difference in simulated
temperature fields.



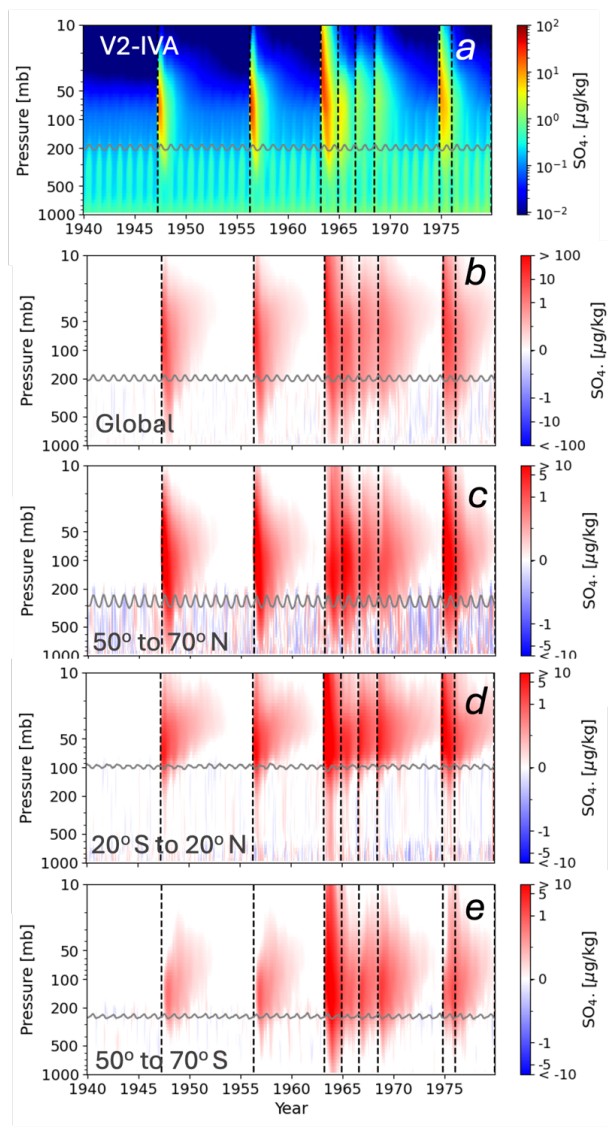

Figure 4. Simulated sulfate (SO4) aerosol concentrations or differences (μg/kg). The x-axis represents time in years, while y-axis represents pressure levels in hPa. The global averaged sulfate aerosol concentrations from V2-IVA are shown in panel *a*. Panels b-e show the SO4 concentration differences between experiment V2-IVA and V2-CMIP6 over different latitude bands. The eruptions recorded by volcanEESM are marked by grey dashed lines (see Table 2).



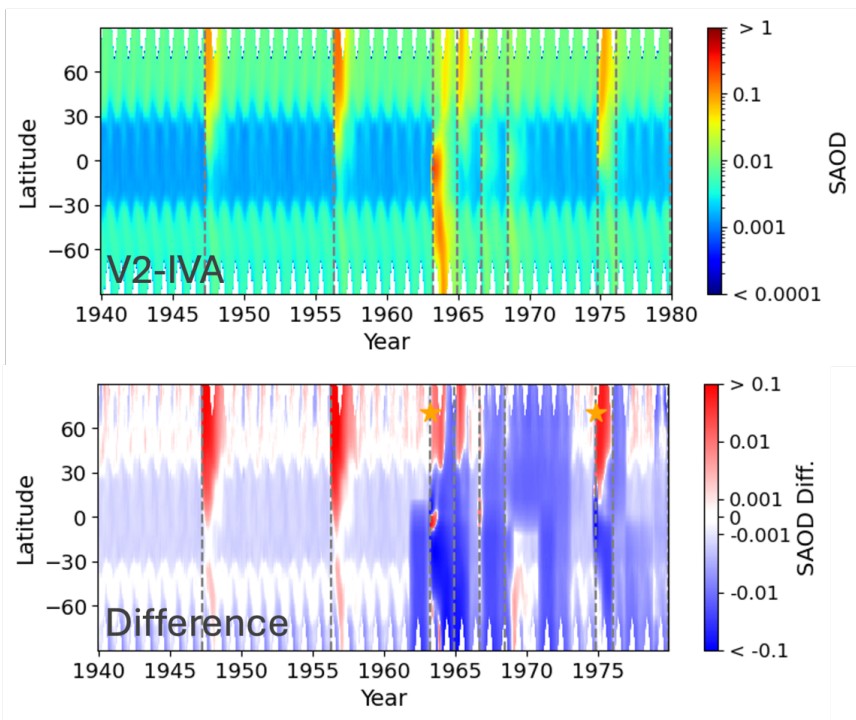

Figure 5. Simulated time (in year) and latitude (in degree) variations of SAOD from V2-IVA ensemble (upper
panel) and the SAOD difference between V2-IVA and V2-CMIP6 (bottom panel). The dashed lines represent
volcanic eruptions in volcanEESM (Table 2), while the stars indicate volcanic eruptions in the CMIP6
documentation (Table 1)



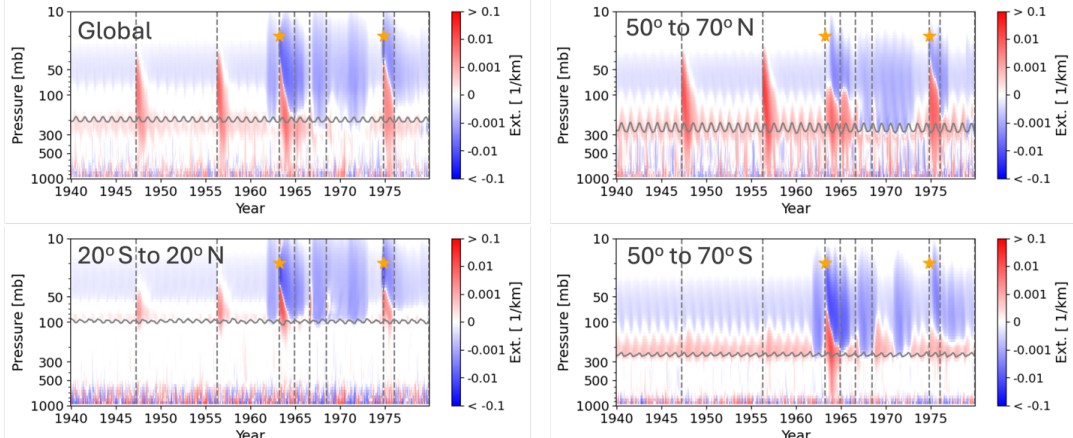

Figure 6. Difference of mean extinction between V2-IVA and V2-CMIP6 over the entire globe and at different
latitude bands. The x-axis represents time in years, and the y-axis represents pressure level in hPa. The vertical
dashed lines represent volcanic eruptions in volcanEESM (Table 2), while the stars indicate volcanic eruptions in the
CMIP6 documentation (Table 1). The solid gray curves represent tropopause simulated by model.



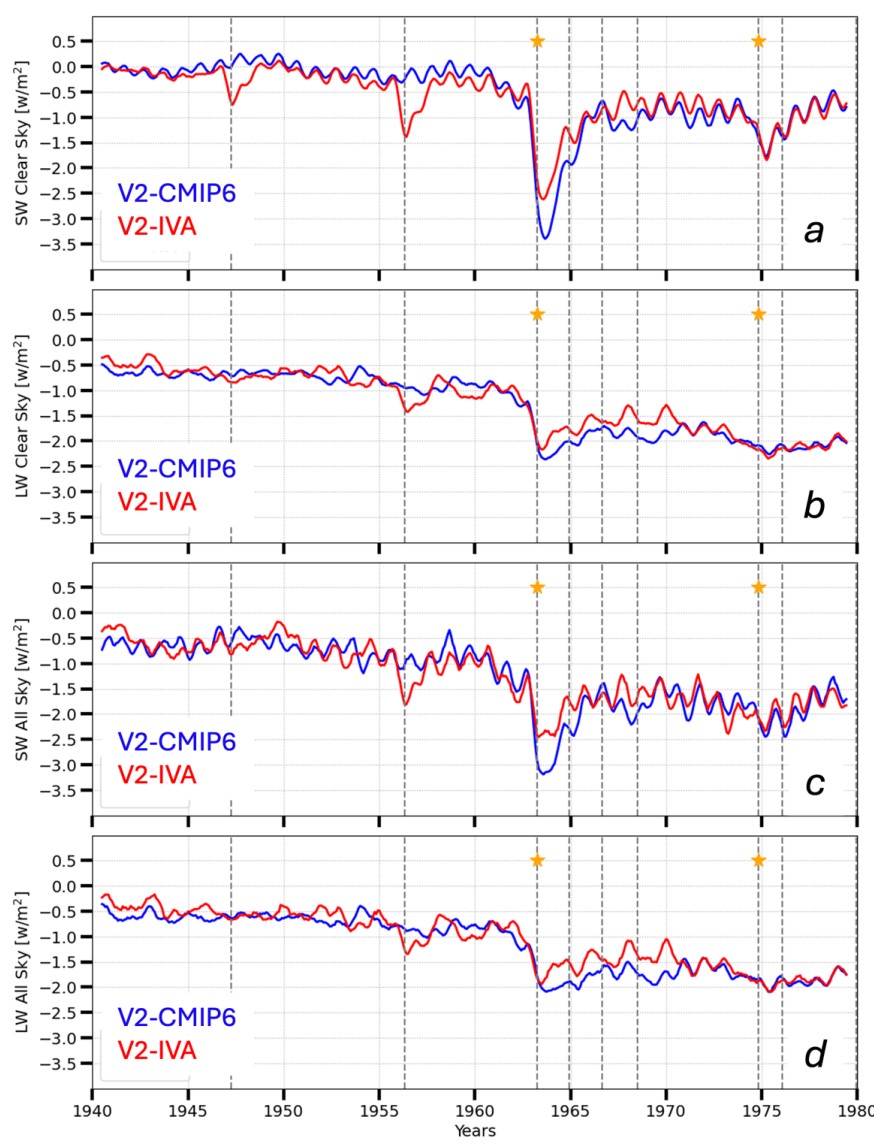

Figure 7. Time series of simulated global mean radiative forcings at the top of model by V2-IVA and V2-CMIP6 ensembles, including the global average of clear-sky shortwave forcing (*a*), clear-sky longwave forcing (*b*), all-sky shortwave forcing (*c*), and all-sky longwave forcing (*d*). The vertical dashed lines represent volcanic eruptions in volcanEESM (Table 2), while the stars indicate volcanic eruptions in the CMIP6 documentation (Table 1).

### 3.2 Simulated Historical Temperature

Since each historical experiment began with a different baseline derived from distinct PI control simulations, directly comparing simulated temperatures across ensembles is not meaningful. Instead, it is more appropriate to compare temperature anomalies from ensemble means, which represent temperature departures from its 1850-1899 climatology. For instance, the temperature anomaly for V2-CMIP6 during the 1940-1979 period was calculated by V2-CMIP6 ensemble mean temperature during year 1940-1979 period subtracting the V2-CMIP6 ensemble





climatology during the 1850-1899 period. This approach intends to evaluate the changes relative to the climate
before anthropogenic emissions took off.
Figure 8 illustrates the difference in temperature anomalies between V2-IVA and V2-CMIP6. During the 1940-1959
period, the eruptions of Helka (1947) and Bezymianny (1956) lead to brief stratospheric warming. However, V2-
IVA shows cooler temperatures in the troposphere shortly after these eruptions compared to V2-CMIP6, particularly
over northern high latitudes. This contrast becomes more pronounced when examining temperature anomalies at
different pressure levels in Figure 9. At the 200 hPa level, V2-IVA exhibites higher temperature anomalies than V2-
CMIP6 after these two eruptions, whereas the situation has been reversed at 500 hPa and the surface. Notably,
eruptions only cause short-lived cooling in the troposphere. In general, V2-IVA simulated a warmer troposphere
than V2-CMIP6 during the 1940-1959 period.
During the 1963-1972 period, dimmer volcanic eruptions resulte in a cooler middle to upper stratosphere in V2-IVA
due to reduced aerosol absorption (Figure 6). Consequently, temperatures at 200 hPa level to the surface are
moderately warmer in the V2-IVA ensemble compared to the V2-CMIP6 ensemble. Figure 9 shows that V2-IVA
simulated temperature anomalies are warmer than V2-CMIP6 simulated ones at all three levels. By 1968, the
temperature difference between V2-IVA and V2 reaches 0.16 °C at the surface, 0.21 °C at 500 hPa, and 0.22 °C at
200 hPa. These findings highlight how differences in the volcanic representation impact interannual temperature
changes. In general, V2-IVA simulates a warmer troposphere than that simulated by V2-CMIP6 during the 1960-
1979 period mainly due to warmer clear-sky shortwave forcing, 0.13 W/m$^2$, resulted from less volcanic aerosol
radiation interaction (ARI) from volcanic eruptions, which agree with previous studies (Chylek et al., 2020).
Previous studies ((Dhomse et al., 2020; Niemeier et al., 2019) indicated that the smaller
Mt Agung (1963) emission, around 7 Tg $SO_2$, should be used in climate models compared to the 9.9 Tg $SO_2$ used in
CMIP6 volcanic forcing simulation (Arfeuille et al., 2014).
Figure 10 shows simulated surface temperature anomalies compared to observations. From 1940 to 1959, unlike
observations, V2-CMIP6 simulates a flat temperature trend, but V2-IVA improves the temperature interannual
variability by incorporating two volcanic eruptions (Helka (1947) and Bezymianny (1956)). This improvement is
reflected in the correlation coefficients between simulations with observation (see Table S1), which increased from
0.15 to 0.38. From 1960 to 1979, V2-CMIP6 simulates a prolonged temperature drop after the Mt. Agung eruption
in 1963 compared to observed temperature trends, while the V2-IVA simulation mitigates this temperature drop. In
the same period, the correlation coefficients of surface temperature between V2-IVA with observation improves to
0.40 compared to the value between V2-CMIP6 with observation. In summary, V2-IVA ensemble with the updated
volcanic emission inventory (volcanEESM) has improved the simulated temperature variability compared to V2-
CMIP6 ensemble.



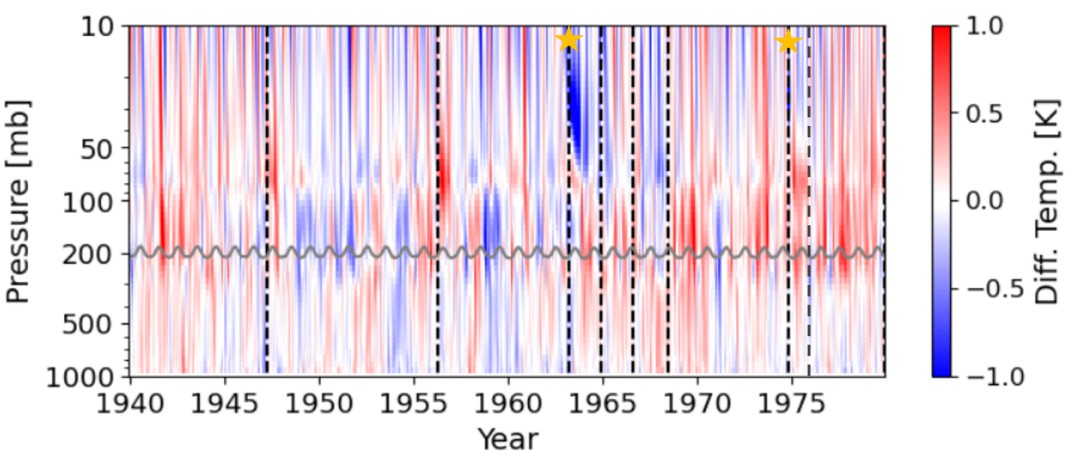

Figure 8. Time-pressure cross-section of global mean temperature difference (K) between V2-IVA and V2-CMIP6 ensembles. The vertical dashed lines represent volcanic eruptions in volcanEESM (Table 2), while the stars indicate volcanic eruptions in the CMIP6 documentation (Table 1). The solid gray curves represent tropopause simulated by model.

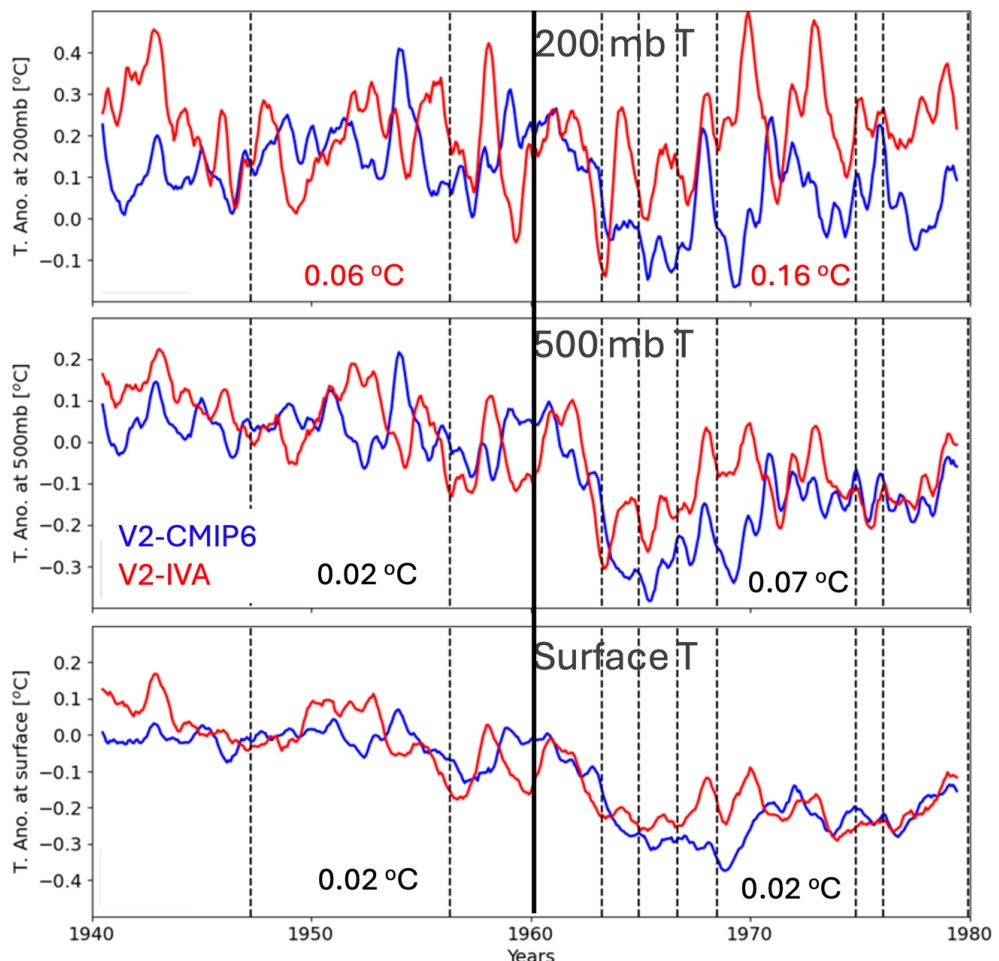

Figure 9. Time series of temperature anomaly at 200 hPa (*top*), 500 hPa (*middle*), and the surface (*bottom*). The dashed lines represent volcanic eruptions in volcanEESM (Table 2). The mean temperature differences during the 1940-1959 period between V2-IVA and V2-CMIP6 are shown in texts at the left side of all panels, while the differences during the 1960-1979 period are shown at the right side of all panels. The number of temperature difference in red color means it is significant with 95% confidence interval.

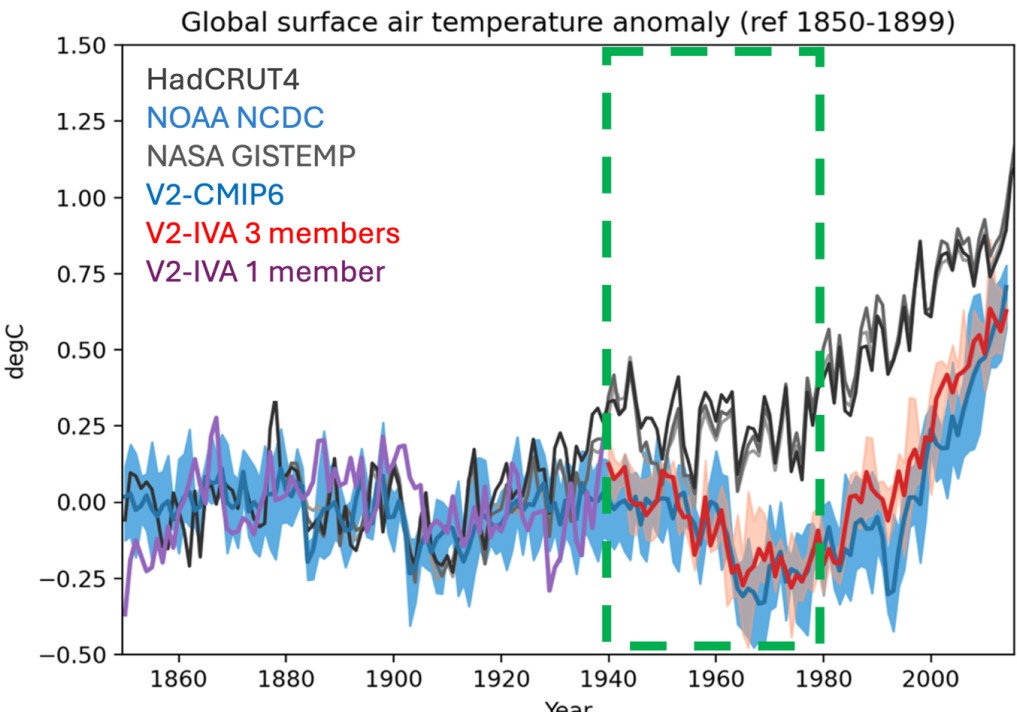

Figure 10. Temperature anomaly trends from 1850 to 2014. Black and gray lines represent observational data, blue
line represents V2-CMIP6 ensemble mean, while red line represent V2-IVA ensemble mean during 1940-1980. The
purple line represents the single member results of V2-IVA from 1850 to 1940.
**3.3 Simulated Decadal Cloud Albedo Forcing Changes**
Cloud forcing change serves as a proxy to depict the strength of aerosol-cloud interactions in the model. Table 4
summarizes the averaged difference of cloud radiative forcing anomalies at the top of the model between V2-IVA
and V2-CMIP6 during the two periods: 1940-1959 and 1960-1979. Additionally, it includes the averaged anomaly
values for V2-IVA and V2-CMIP6 in brackets. For example, during the 1940-1959 period, the net cloud forcing
anomalies simulated by the V2-CMIP6 experiment was -0.72 W/m², indicating that the cloud forcing leads to a 0.72
W/m² cooling effect compared to its 1850-1899 climatology. This cooling is primarily attributed to increased
anthropogenic emissions during this period, leading to enhanced aerosol-cloud interactions and consequently
positive cloud fraction anomaly (see Table 4, total cloud fraction), supported by low cloud fraction anomaly
increase. With further increases in anthropogenic emissions during the 1960-1979 period, the cloud forcing anomaly
has increased to a greater magnitude (more negative), adding another 0.19 W/m² to -0.91 W/m². The cloud forcing
values are in line with Golaz et al. (2022) and comparable to values from other models during the similar period
(Bauer et al., 2020; Flynn and Mauritsen, 2020; Zhang et al., 2021). This contributes to the simulated surface
temperature low bias by E3SMv2 (Golaz et al., 2022). Notably, since there are no eruptive volcanic aerosols in V2-
CMIP6, volcanic aerosols have no effect on the V2-CMIP6 simulated cloud forcing anomalies through aerosol-
cloud interactions.



In the V2-IVA experiment, prognostic volcanic aerosols (specifically sulfate in this study) are involved in aerosol-
cloud interactions. Additionally, the volcanic aerosols' historical averaged emissions has been incorporated into the
PI control simulations. Thus, historical changes of the volcanic emissions affect cloud formation and cloud radiative
forcing. During the 1940-1959 period, the volcanEESM inventory recorded two eruptions, Helka (1947) and
Bezymianny (1956), emitting a total of 6.2 Tg $SO_2$ into the atmosphere, equivalent to an average emission of 3.1 Tg
per decade. This value is notably smaller than both the historical average (7.0 Tg per decade) and the 1850-1899
climatology (6.5 Tg per decade), indicating a reduction in volcanic $SO_2$ emissions during this period compared to
the earlier climatology. This reduction could partially offset the growth in anthropogenic emissions, resulting in a
warmer climate compared to the V2-CMIP6 experiment (volcanic quiescent warming effect).
During the 1940-1959 period, V2-IVA has simulated net cloud forcing anomaly were -0.61 W/m², which is 0.11
W/m² warmer than that of V2-CMIP6, representing a 15% reduction of cloud cooling compared to V2-CMIP6. This
warming is caused by a 0.16 W/m² reduction in shortwave cloud forcing warming. Table 4 provides a breakdown of
cloud property changes, revealing that the V2-IVA simulated total cloud fraction anomaly is 52% less than the value
simulated by V2-CMIP6. This is mainly due to the much smaller low cloud fraction anomalies in V2-IVA
simulation, compared to that in V2-CMIP6 simulation. Consequently, the total grid-mean liquid water path anomaly
is 7% less than the value in V2-CMIP6. These changes are statistically significant at the 95% confidence level. As a
result, V2-IVA simulated temperature anomaly is warmer than the V2-CMIP6 simulated value, although these
differences may not pass the significance testing as various factors can contribute to temperature fluctuations (
582 9).

In contrast, there were 16.4 Tg $SO_2$ emissions from volcanic eruptions during the 1960-1979 period, equivalent to
8.2 Tg per decade (see Table 2), which exceeded the 1850-1899 climatology of 6.5 Tg per decade. Consequently,
the additional emissions resulted in a 28% increase of low cloud fraction anomaly and a 5% increase of liquid water
path anomaly, comparing V2-IVA with V2-CMIP6. This leads to a cooling effect on net cloud forcing at -0.08
W/m² (see Table 4). This is considered as the volcanic surplus cooling effect. However, it's important to note that
these differences hasn't passed significance testing, potentially due to the relatively large number of anthropogenic
emissions during the same period, which masks the impact of the volcanic aerosols. Interestingly, despite the -0.08
W/m² cooling effect on net cloud forcing, the simulated temperature anomalies in experiment V2-IVA were slightly
warmer than those in the V2-CMIP6 experiment (see Figure 9). This warming can be mainly attributed to the
alleviated volcanic aerosol direct forcing, which resulted in a warming of 0.13 W/m² under clear-sky conditions. The
simulated cloud fraction anomaly values are reasonable and in line with the study that evaluates CMIP6 cloud
fraction variations across different climate models (Vignesh et al., 2020).
In summary, the aerosol-cloud interactions induced by volcanic aerosols fluctuate upon changes in volcanic eruption
magnitudes. During the volcanic quiescent periods, such as 1940-1959, interactive volcanic aerosol representation
mitigates the cooling effect from cloud radiative forcing by offsetting the increase of the anthropogenic aerosols.
Conversely, during the volcanic active periods, the new interactive treatment intensifies the cooling effect by
introducing more sulfate into the atmosphere. This finding provides new insight into understanding the surface
temperature low bias spreading across climate models (Flynn and Mauritsen, 2020; Zhang et al., 2021).



Table 4. Differences in global mean cloud properties between V2-IVA and V2-CMIP6. The numbers in the brackets
are anomaly resulted from V2-IVA and V2-CMIP6 ensembles, respectively. Red shaded values represent
statistically significant at the 95% confidence level.

|  | 1940-1959 | 1960-1979 |
|---|---|---|
| Net Cloud Forcing (W/m$^2$) | 0.11 (-0.61, -0.72) | -0.08 (-0.99, -0.91) |
| SW Cloud Forcing (W/m$^2$) | 0.16 (-0.50, -0.65) | -0.04 (-0.78, -0.74) |
| Total Cloud Fraction (%, grid mean) | -0.146 (0.137, 0.283) | -0.005 (0.306, 0.311) |
| Low Cloud Fraction (%, grid mean) | 0.113 (0.131, 0.244) | 0.081 (0.369, 0.289) |
| Cloud Liquid Water Path ($10^{-5}$ * kg/m$^2$, grid mean) | -7 (97, 105) | 7 (148, 141) |

**3.4  The Effect of Volcanic Aerosols in PI Control**
The PI control simulation is designed to establish the baseline climate for historical transient simulations (Schmidt et
al., 2012). In this study, V2-IVA-NPI ensemble experiments are conducted to explore the effect of volcanic aerosols
in the PI-control configuration on simulated historical climate (Table 3). The hypothesis is that without including
historical averaged volcanic aerosols in the PI control, the additional sulfate aerosol emissions from historical
volcanic eruptions would contribute to enhancing aerosol-cloud interactions and cooling of the climate.
Additionally, without the inclusion of volcanic aerosols in the PI control, the volcanic quiescent warming and
surplus cooling effects cannot be represented. The V2-IVA-NPI experiment replicates the setup of the V2-IVA
experiment, but it omits the historical averaged explosive volcanic aerosols in the V2-IVA-NPI's PI control run,
which serves as the initial condition for the V2-IVA-NPI's historical run. Although both V2-IVA and V2-IVA-NPI
experiments have the identical emissions in the historical run, we anticipate that the anomaly in V2-IVA-NPI,
relatives to its 1850-1899 climatology, result in more low clouds, an enhancement of cooling via aerosol-cloud
interactions, and a cooler climate compared to its V2-IVA counterpart.
Table S2 presents a comparison between the V2-IVA and V2-IVA-NPI experiments. During the 1940-1959 period,
both V2-IVA and V2-IVA-NPI show an increase in low cloud anomaly compared to their 1850-1899 climatology.
Notably, the increase in low cloud fraction in V2-IVA-NPI has been significantly higher than that in V2-IVA.
Additionally, the V2-IVA-NPI simulation has simulated a larger liquid water path anomaly, compared to that in V2-
IVA. Consequently, the net cloud forcing anomaly in V2-IVA-NPI is 0.10 W/m² cooler than that in V2-IVA,
indicating a cooling from aerosol-cloud interactions. Furthermore, the V2-IVA-NPI simulation exhibits a
temperature anomaly that is 0.09 K cooler than that in V2-IVA, which is statistically significant. A similar, albeit
less pronounced, pattern is observed during the 1960-1979 period, with V2-IVA-NPI simulating a cooler climate
compared to that in V2-IVA. In general, the results are qualitatively agree with previous study about the importance
of the volcanic aerosols in PI control simulation (Chim et al., 2023; Schmidt et al., 2012).
These findings highlight the significant role of volcanic aerosols in shaping historical climate simulations and
emphasize the importance of their inclusion in climate models' baseline simulations.
**4.  Conclusions and Discussion**
This study investigates whether the representation of volcanic eruptions in E3SM could be contributing to its low
temperature bias during 1940-1980 and evaluates the impact of volcanic representations on the simulated climate.
The standard E3SMv2 model, following the CMIP6 protocol, represents volcanic eruptions by prescribing
simplified radiative forcing and neglects the interactions between volcanic aerosols and clouds. Instead, in this study





we introduce another representation that treats volcanic eruptions as $SO_2$ gas emissions and the induced sulfate
aerosols in aerosol processes using MAM4 to represent the volcanic aerosol-radiation and aerosol-cloud
interactions.
The experiments consist of a control run, V2-CMIP6 ensemble and two test runs, V2-IVA and V2-IVA-NPI
ensembles. The control run utilizes historical transient simulations with five members from 1850 to 2014. The V2-
IVA ensemble includes a 100-year spin-up under the same PI control configuration as the control run but using the
averaged volcanic emissions from 1850 to 2014 as the background eruptive volcanic emission. After the spin-up,
one member is simulated from 1850 to 2014, with two additional members added from 1940 to 2014 to reduce the
interannual variability in coupled simulations. The V2-IVA-NPI ensemble is identical to V2-IVA ensemble, but
without any background explosive volcanic emissions in its 100-year PI control.
Our analysis indicates that while the improvement in volcanic aerosol representation does not profoundly alter
model outcomes, there are noticeable improvements, albeit at relatively small magnitudes. The V2-IVA ensemble,
which utilizes the latest volcanEESM inventory includes previously unaccounted eruptions like Helka (1947) and
Bezymianny (1956), alongside an adjusted representation of Mt. Agung's eruption (1963). It exhibits enhancements
in the simulated surface temperature temporal variability. Specifically, the surface temperature anomaly correlation
coefficient between observation and model simulated results is increased from 0.15 to 0.39 during the 1940-1959
period and from 0.32 to 0.40 during the 1960-1979 period. Additionally, the V2-IVA simulated atmosphere is
marginally warmer (by 0.02 to 0.15 K) across various pressure levels (from 200 hPa to the surface) compared to the
V2-CMIP6 atmosphere (Figure 9).
Furthermore, our study unveils the mechanisms by which historical volcanic eruptions affect cloud forcing
variability. During volcanic quiescent periods, characterized by eruptions below the historical average, a reduction
in volcanic emissions partially offsets the increase of anthropogenic emissions, resulting in a volcanic quiescent
warming effect. Conversely, during volcanic active periods, excessive volcanic emissions augment anthropogenic
emissions with more sulfate aerosols, leading to a volcanic surplus cooling effect. These effects are evident when
comparing cloud forcing anomalies between the V2-IVA and V2-CMIP6 simulations during the 1940-1959 and
1960-1979 periods (see Table 4). Specifically, during the 1940-1959 period, characterized by volcanic quiescence,
the net cloud forcing anomaly in the V2-IVA simulation is 0.10 W/m² warmer than that in the V2-CMIP6
simulation. In contrast, during the 1960-1979 period, characterized by volcanic emission amount surplus historical
mean, the net cloud forcing anomaly in the V2-IVA simulation is 0.07 W/m² cooler than that in the V2-CMIP6
simulation. These changes in cloud forcing are evidently reflected in the low cloud fraction anomaly and liquid
water path anomaly comparisons. This finding gives new insight to understanding the surface temperature low bias
spread across many climate models (Flynn and Mauritsen, 2020; Golaz et al., 2022; Zhang et al., 2021).
It is worth noting that this study uses the MAM4 to represent volcanic sulfate aerosols, which needs some
improvements to accurately reproduce Mt. Pinatubo (1991) eruption (Mills et al., 2014). This effort will be
represented in a following-up paper that documents a new development of adding a stratospheric sulfate mode on
top of MAM4 for E3SM version 3 (Ke et al., in preparation). However, the goal of this study is to highlight the
improved model performance when the representation of variability in volcanic aerosols is improved: an interactive
volcanic aerosol treatment and an updated volcanic emission inventory. Importantly, the volcanic quiescent warming
effect introduced by this study potentially plays an important role in the rapid global warming during the 1920-1960
period, during which eruptive volcanic emission amount was 2.8 Tg per decade, much lower than the historical
average of 7.1 Tg per decade. This warming trend known as early twentieth-century warming in observations but is
missed in climate model simulations (Brönnimann, 2009; Hegerl et al., 2018).
Future research is warranted to focus on improving the volcanic aerosols' representation in climate models, such as
the size distribution, mixing (with other aerosol components), activation, and ice nucleation processes.



***Code and data availability***
The dataset had been analyzed in this study are available at https://doi.org/10.5281/zenodo.11246313, E3SMv2
source code at https://doi.org/10.5281/zenodo.11403736 and E3SMv2 run script at
https://zenodo.org/records/11403988 The E3SM project, code, simulation configurations, model output and tools to
work with the output are described on the E3SM website (https://e3sm.org, last access: 20 May 2024). Instructions
on how to get started running E3SM and its components are available on the E3SM website
(https://e3sm.org/model/running-e3sm/e3sm-quick-start, last access: 20 May 2024).
***Competing interests***
At least one of the (co-)authors is a member of the editorial board of Geoscientific Model Development
***Author contribution***
All co-authors designed the experiments and Ziming Ke carried them out V2-IVA and V2-IVA-NPI experiments.
The V2-CMIP6 results had been provided by Jean-Christophe upon Golaz et al. (2022) study. Ziming Ke performed
the data analysis and all co-authors provided contributions. Ziming Ke prepared the manuscript with Xiaohong Liu
provided significant revisions. All co-authors contributed to final manuscript revisions.

***Acknowledgments***
This research was supported as part of the Energy Exascale Earth System Model (E3SM) project, funded by the US
Department of Energy (DOE), Office of Science, Office of Biological and Environmental Research (BER).
Lawrence Livermore National Laboratory (LLNL) is operated by Lawrence Livermore National Security, LLC, for
the US DOE, National Nuclear Security Administration under contract no. DEAC52-07NA27344. Support was
received from the LLNL LDRD project 22-ERD-008, "Multiscale Wildfire Simulation Framework and Remote
Sensing".
The Pacific Northwest National Laboratory (PNNL) is operated for DOE by Battelle Memorial Institute under
contract DE-AC05-76RLO1830.
We thank Paul J. Durack and Jiwoo Lee from LLNL for helping with manuscript discussion.
The data was produced using a high-performance computing cluster provided by the BER Earth System Modeling
program and operated by the Laboratory Computing Resource Center at Argonne National Laboratory.
The E3SM version 2.0 have been used in this study. The source code can be found at this link
(https://github.com/E3SM-Project/E3SM/releases/tag/v2.0.0. 29 Sep. 2021. Web.
doi:10.11578/E3SM/dc.20210927.1)



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
