# Peer review of "Assessing the Climate Impact of an Improved Volcanic Sulfate Aerosol Representation in E3SM"

_EGUsphere, 2024_

## Author Comment (AC1)

This study assesses the role of volcanic aerosols on 20th century climate, in particular whether simulating volcanic aerosol interactions with clouds (and a few added small explosive eruptions) leads to improved historical climate replication. Global climate model (GCM) treatments of volcanic aerosols are typically highly simplified, with impacts of interactive volcanic aerosol processes being rarely accessed for more than a decade. There has been a lot of recent effort with E3SM to assess potential improvements in climate simulations resulting from interactive aerosol and cloud process additions, and these volcanic aerosol impacts are a worthwhile candidate. The experiments in this study are a fitting and worthwhile effort, though there are considerable issues with how the results are presented. Most critically, the manuscript claims substantial improvement but does not clearly show this to be the case. The vast majority of the targeted historical temperature bias remains despite the altered volcanic scenario, so it looks clear that improvements are small. These results, though mostly negative, are of interest to other modeling centers and worth ultimately being published. But I must ask the authors to please accurately communicate the results, to add figures to show whether the hypotheses hold any water (i.e. volcanic impacts on CCN, net all-sky and cloud forcings), and to choose a more accurate title.

Our responses are in blue color:

We sincerely appreciate the reviewer's thorough evaluation and constructive feedback. We acknowledge the concerns regarding the simplified treatment of volcanic aerosols in global climate models (GCMs) and the importance of assessing the long-term impacts of interactive volcanic aerosol processes. As the reviewer correctly pointed out, these processes have rarely been evaluated over multi-decadal timescales, making this study a valuable contribution to understanding their role in historical climate simulations.

In our original manuscript, we overstated the impact of eruptive volcanic aerosols on simulated surface temperature. To address this, we have revised the title to "**Assessing the Climate Impact of an Improved Volcanic Aerosol Representation in E3SM**." Additionally, our updated analysis clarifies that while our study provides a more physically consistent representation of volcanic eruptions in E3SM, the improvements in simulated surface temperature are marginal rather than a substantial reduction in temperature bias. The revised text now more accurately reflects these findings.

As requested, we have expanded our analysis to quantify the effects of volcanic aerosols on cloud droplet number concentration (CDNUMC), cloud cover, cloud radiative forcing, and net radiative forcing at both the top of the atmosphere and the surface. These additions strengthen our conclusion that eruptive volcanic aerosols influence the radiative forcing budget and alter the simulated climate through aerosol-cloud interactions. Additionally, we have revised the **Discussion** section of the manuscript to provide a more detailed analysis of how volcanic aerosols influence cloud processes in E3SM and other climate models. We also discuss potential improvements in aerosol and cloud representations that could enhance future simulations.

Major Comments

The title is very vague and optimistic to an extent not supported by the results. This work is about the role of specific volcanic aerosol processes in historical climate simulations, so should be titled along these lines. E.g. 'Impact of explosive volcanic eruptions on mid-20th century surface temperature development: the role of aerosol-cloud interactions' or something similarly explicit. Climate model performance is only minimally shown to be "enhanced", as the lion's share of historical temperature bias remains, and "performance" is unclear.

We appreciate the reviewer's feedback and agree that the original title was too vague and optimistic. We have revised it to:

**"Assessing the Climate Impact of an Improved Volcanic Aerosol Representation in E3SM"**

While the simulation changes are appropriate, the key claim is that in Fig. 10 the altered experiment (red line) is more similar to observations (black and grey) than the original experiment (dark blue line), and that this is due to more accurate volcanic representation. But I do not find the improvement or its cause clear, so must ask the authors to address this in two ways. First, I want to see that the net TOA forcing evolution is clearly different between the model versions, as a substantial net forcing difference is requisite for a substantial forcing-driven surface temperature anomaly, and the first is less susceptible to the very heavy noise in Fig. 10. So I request that net forcings be shown in Figure 7, along with appropriate uncertainty bounds of the authors choosing. Second, because one of the main hypotheses is that the new setup improves climate by accurately representing aerosol-cloud interactions, I want to see evidence of this: First, some depiction of CCN number density differences between experiments, and second a depiction of net cloud radiative effect/forcing (CRE) development over time.

We appreciate the reviewer's detailed feedback and suggestions for improving the clarity of our results. To address these concerns, we have made several revisions. To better illustrate the differences in radiative forcing between model versions, we have added the net top-of-atmosphere (TOA) radiative forcing evolution to **Figure 7**, along with appropriate uncertainty bounds. This provides a clearer comparison of how forcing differences drive surface temperature anomalies and offers a less noisy metric than surface temperature variations shown in **Figure 10**.

Additionally, to support our hypothesis that improved volcanic aerosol representation enhances climate simulations through aerosol-cloud interactions, we have included an analysis of vertical integrated **cloud droplet number concentration (CDNUMC) differences** between experiments. Instead of cloud condensation nuclei (CCN) number density, we use CDNUMC as

it provides a more direct indicator of cloud microphysical responses to volcanic aerosols, particularly in the context of aerosol-cloud interactions affecting cloud properties and radiative forcing. To further support this, we also added a depiction of **net cloud radiative forcing over time**, which provides a clearer link between aerosol changes and their radiative impacts.

The results appear quite negative yet the manuscript is extremely upbeat about large improvement, which I find deceptive. I think the results being somewhat negative is interesting, and that this is overall a worthwhile effort, so the strongly upbeat tone is confusing and unnecessary. The model changes do not come close to fixing the targeted bias, with the revised model's global mean surface temperatures veering off into practically the same bias as the original model around the time of the Agung eruption. Yet the manuscript is in total denial of this, cf Lines 505-6: "V2-CMIP6 simulates a prolonged temperature drop after the Mt. Agung eruption in 1963 compared to observed temperature trends, while the V2-IVA simulation mitigates this temperature drop." If the requested figure panels work out as anticipated, the authors can tout a small-to-moderate amount of improvement, but the results and conclusions can't grossly misrepresent the data. Further, this denial prevents the authors from explaining why the results are not stronger, which would be interesting and beneficial to know.

We appreciate the reviewer's feedback and have revised the manuscript to ensure a more measured discussion of our results. While our improved volcanic aerosol representation leads to some mitigation of the Mt. Agung 1963 temperature drop, the overall bias remains substantial. We have adjusted the tone throughout the results and conclusions sections to avoid overstating improvements and have reworded **Lines 505–506** accordingly. Additionally, we appreciate the reviewer's suggestion to discuss **why the improvements are limited**, including potential biases in **overestimate of aerosol burden, simplified chemistry, lack of updated wet removal process, and lack of historical changes of tropospheric ozone**. These factors may contribute to the remaining discrepancies, and we now address them in the revised discussion.

The manuscript is often confusing to read and repeats itself, so can be shortened and better organized. Section 2.3 is largely a repeat of the latter half of the Introduction that seems unnecessary. Similarly, the Discussion is mostly a repeat of the Results rather than a discussion of caveats, lessons for future efforts, etc. The section on forcings from aerosol-cloud interactions (3.3) would more logically go before the temperature anomalies these are presumed to cause (3.2). I also feel the preindustrial control simulations are too disjoint from the central results about temperature changes to warrant their own results section (3.4), but that some of this material can be used elsewhere. Further, several aspects of this study, which I've laid out in the specific comments, make it confusing to understand and should to be clearer.

We appreciate the reviewer's suggestions on improving the clarity and structure of the manuscript. To address these concerns, we have removed **Section 2.3** and integrated its content into the **Introduction** to avoid redundancy. The **Discussion section** has been revised to focus more on caveats, lessons for future work, and broader implications rather than reiterating the Results. To enhance logical flow, we have reordered the manuscript by moving the **aerosol-cloud interaction forcings** section before the **temperature anomalies** discussion. Additionally, the **preindustrial control simulations** have been incorporated into relevant sections to better connect them with the study's main findings. We have also refined key aspects of the study based on the reviewer's specific comments. We appreciate this constructive feedback and believe these changes improve the manuscript's clarity and organization.

Specific Comments
Line 13: The abstract's first sentence is odd for two reasons: 1) The phrase "representation of Earth's surface temperature" is vague, as accurately simulating historical temperature is what presumably lends credence to future projections, rather than simply representing present-day climatological temperatures, and 2) The revised simulation setup still has considerable bias in replicating historic temperature, so if this would be an argument *against* using E3SM for climate projections.

We appreciate the reviewer's suggestions. We have reworded the opening sentence to emphasize the importance of accurately simulating historical surface temperature variations for evaluating climate models. This revision avoids implying that our study fully resolves biases or directly validates E3SM's projections.

Line 23: It's pretty impossible to know what the "volcanic quiescent warming effect" is before it is properly introduced, so I hope the authors can describe this in simpler language here, e.g. removal of artifacts that stem from a simplified volcanic background aerosol state.
We appreciate the reviewer's suggestions. We recognize that this term was unclear. In the revised abstract, we now describe this effect as **"aerosol-cloud interactions reducing cloud forcing cooling during volcanically low activity periods,"** providing a more explicit and accessible explanation.

Line 25: "strongly in favor" would indicate the influence on climate is strong, which is not the case. The wording in the abstract is too positive given the largely negative results.

We acknowledge that the original wording overstated the impact of our findings. We rewrite the abstract as:

Accurately simulating historical surface temperature variations is essential for evaluating climate models, yet many struggle to reproduce the mid-20th-century temperature trends associated with significant volcanic eruptions. This study examines the impact of volcanic sulfate aerosol representation on these biases using the Energy Exascale Earth System Model (E3SM). The

standard CMIP6 protocol prescribes volcanic forcing through radiative perturbations, omitting volcanic aerosol-cloud interactions (VACIs). Here, we implement an emission-based approach with an updated volcanic eruption inventory that directly incorporates volcanic sulfur dioxide ($SO_2$) emissions, enabling a more process-based representation of volcanic forcing. This approach leads to improved surface temperature variability and a modest reduction in cold biases between 1940 and 1980 compared to the CMIP6 setup. Additionally, we assess cloud property responses to a more realistic volcanic sulfate aerosol representation, which weakens cloud-induced cooling during periods of lower volcanic activity. However, despite these refinements, a significant temperature cold bias remains, indicating that further improvements in aerosol microphysics, cloud processes, and model parameterizations are needed to fully resolve this issue in E3SM.

Lines 34-6: I feel the first paragraph of the Intro would be stronger if it kept to the focus of this study, rather than alluding to much stronger eruptions. The focus here is on mid-20th century eruption impacts, of which Agung's 0.1-0.2 C cooling is the most consequential. I hence find the Tambora and Pinatubo references distracting, as they're multiple times stronger than any event in the 1940-80 assessed window. Likewise, the line that eruptions play "a crucial role in modulating climate changes" most evokes impacts of greenhouse gases from Large Igneous Provinces, as "crucial" is debatable for pretty much any other eruption and certainly the relatively modest ones assessed in this study. I hope the authors can make this study's focus interesting rather than emulate standard intros of volcano-climate literature. The authors are free to do as they want here, though one path could be to start by mentioning that climate models have trouble replicating 20th century surface temperature evolution, that this reduces confidence in projections of future climate, the entangled roles of anthropogenic aerosol, greenhouse gas, and natural aerosol changes, introducing why poor representation of volcanic eruptions is a primary suspect, and briefly describing Agung's cooling as the largest volcanic impact of this period.

We appreciate the reviewer's feedback and understand the concern about maintaining the study's focus on mid-20th-century eruptions, particularly Mt. Agung's impact. However, we believe that briefly mentioning Tambora and Pinatubo provides important context for a broader audience who may not be familiar with the climatic effects of volcanic eruptions. Tambora and Pinatubo are widely recognized as historically significant eruptions that illustrate the role of volcanic activity in climate modulation—Tambora marking a major preindustrial event and Pinatubo serving as a key reference in the satellite era. While Mt. Agung's eruption is comparable to Pinatubo, the latter is more well-known, making it a useful reference point for readers.

Lines 34: It seems odd to start with "natural radiative forcing" before saying that volcanic eruptions result in sunlight-blocking aerosols. I'd suggest holding off on this term until it is defined.

We agree with this suggestion and delete corresponding phrases.

Line 45: The current placement of "Water vapor is scarce in the stratosphere" after a line saying that "eruptions emit a variety of gases [...] into the stratosphere" suggests this study will involve water vapor emissions into the stratosphere, but this gets no further mention. I would weave this into the following line to remove the perceived focus on water vapor, ie "lack of wet removal in the water-scarce stratosphere".

We appreciate this suggestion and revised sentence as "As lack of wet removal in the water-scarce stratosphere, sulfate aerosols can persist for months to years due to lack of wet removal as compared to days in the troposphere"

Line 53: Because this study's main focus is on aerosol-cloud interactions, it would be fitting for there to be a paragraph here on whether past research has indicated a strong effect or not. For instance there are the two studies I cite below, one claiming 'substantial cooling', the other a weak effect (oddly with largely the same authors). I suspect there are other relevant studies. It would also be beneficial to have a little overview of arguments for why the effect could be weak and why it could be strong, e.g. whether volcanic sulfate falls to the troposphere over wide areas or mostly near the poles, the global abundance of CCN, scalings of cloud optical depth and radiative forcing with CCN number density). I also hope the authors can clarify if this study's focus is solely on explosive eruptions, or if effusive eruptions – which also release sulfate and hence affect cloud properties – are also a focus.
Chen, Y., Haywood, J., Wang, Y., Malavelle, F., Jordan, G., Peace, A., ... & Lohmann, U. (2024). Substantial cooling effect from aerosol-induced increase in tropical marine cloud cover. Nature Geoscience, 1-7.
Malavelle, F. F., Haywood, J. M., Jones, A., Gettelman, A., Clarisse, L., Bauduin, S., ... & Thordarson, T. (2017). Strong constraints on aerosol–cloud interactions from volcanic eruptions. Nature, 546(7659), 485-491.

We appreciate the reviewer's suggestion to include a discussion on past research regarding aerosol-cloud interactions from volcanic eruptions. We have added a few sentences summarizing previous findings and discussing potential reasons for the variability in observed effects. In this study, we focus on explosive eruptions.

"The impact of volcanic aerosols on cloud properties depends significantly on background aerosol conditions and regional meteorology. Chen et al. (2024) found that volcanic eruptions in Hawaii led to a significant increase in cloud cover, enhancing reflected sunlight and contributing to a substantial cooling effect. In contrast, Malavelle et al. (2017) examined volcanic aerosols near Iceland and found that while cloud brightness increased, changes in cloud cover and liquid cloud water were minimal, leading to a weaker overall effect. These differences highlight the complex interplay between volcanic sulfate injection, cloud condensation nuclei (CCN) availability, and regional cloud microphysics."

Line 55 – Flynn & Mauritsen 2020 is cited 4x in this manuscript, and its Fig. 12 is quite relevant here. But I hope the authors appreciate that the aerosol-cloud interactions that study raises as the cause of CMIP6 historical climate bias are influences of ice nuclei on mixed-phase cloud phase, so extremely different than the interactions assessed here. Hence I find the wording here a little misleading, though it's vague enough I find it okay if the authors keep it.

We appreciate the reviewer's clarification regarding the aerosol-cloud interactions discussed in Flynn & Mauritsen (2020). While their study primarily attributes the increase in Earth's equilibrium climate sensitivity (ECS) from CMIP5 to CMIP6 to changes in mixed-phase cloud processes and Antarctic sea ice representation, we acknowledge that ECS is not the focus of our study.

However, Flynn & Mauritsen (2020) also highlight that the transient climate response (TCR) in CMIP6 models is cooler than in CMIP5, largely due to stronger aerosol cooling, which results in an underestimated mid-20th-century warming compared to instrumental records. Since our study investigates the role of volcanic aerosol-cloud interactions in CMIP6 temperature biases, particularly in the mid-20th century, the reference remains relevant to our discussion.

We have carefully reviewed our wording to ensure it accurately reflects the findings of Flynn & Mauritsen (2020) without implying a direct connection to mixed-phase cloud processes. We appreciate the reviewer's attention to this distinction.

Lines 62-67: Moving the first 4 sentences of this paragraph into the previous paragraph would allow this one to focus specifically on explaining the two mechanisms. As these are tricky to understand, keeping the paragraph focused would help the reader.

We appreciate this suggestion and moved the revised four sentences of the paragraph into a separated paragraph.

Lines 67-76: I find this critically important paragraph confusing so request the authors please rewrite this. First, the "volcanic quiescent warming" name is confusing, as there is no physical volcanic warming process here and this seems to be more of a "reduced volcanic cooling during quiescent periods" scenario due to removal of a simplified process representation (constant background volc aerosol). This should at minimum be explained more clearly. Second, the discussion of anthropogenic aerosol changes here is confusing to me, as the two mechanisms both involve altered volcanic aerosols in simulations with the same anthropogenic aerosol changes. Interactions between volcanic and anthropogenic aerosol processes can certainly be an influence, but I don't see this as an explanation for either mechanism, so think it makes more sense to focus on just the volcanic aerosol changes.

We appreciate the reviewer's feedback and have rewritten the paragraph to clarify the mechanisms without using the terms "volcanic quiescent warming effect" and "volcanic surplus cooling effect," which may have been misleading. Instead, we now explain that including volcanic aerosols in the pre-industrial background creates a more realistic baseline, where cloud radiative forcing varies depending on volcanic activity levels. Additionally, we have refined the discussion to ensure that anthropogenic aerosol effects are not misleadingly presented as part of these mechanisms. We now focus solely on volcanic aerosol processes, aligning with the reviewer's suggestion.

"The CMIP6 protocol represents volcanic sulfate aerosol-radiation interactions in the stratosphere using prescribed stratospheric forcing (Figure 1, left panel). During the satellite era, this approach blends multi-satellite observations, providing relatively accurate representations of VARIs. However, it neglects volcanic sulfate aerosol-cloud interactions, which can significantly influence cloud properties and radiative forcing. To address this limitation, we implement an emission-based approach, replacing prescribed volcanic stratospheric forcing with volcanic $SO_2$ emissions (Figure 1, right panel). This method, which captures both VARIs and VACIs, has been used in previous studies (e.g., Brown et al., 2024; Mills et al., 2016).

To improve model initialization, we incorporate historical average volcanic $SO_2$ emissions into pre-industrial (PI) control simulations, establishing a different baseine for subsequent historical transient simulations (1850–2014). CMIP6 guidelines recommend including averaged natural forcing into PI control simulations (Eyring et al., 2016). Schmidt et al. (2012) demonstrated that adding natural aerosols in PI and subsequent historical transient simulations could potentially dampen the magnitude of the increase of aerosol-cloud interactions in the historical transient period. Without volcanic sulfate aerosols, the ratio ($R_0$) of historical aerosol increases ($A_a$), which is mainly driven by anthropogenic emission, relative to the PI average level ($A_{PI}$) is expressed as equation (1).

$$R_0 = \frac{A_a}{A_{PI}} \qquad (1)$$

When volcanic sulfate aerosols ($A_v$) are included, the new historical aerosol increase ratio, $R_v$, can be expressed as equation (2), where $\overline{A_v}$ is historical average of volcanic sulfate aerosols.

$$R_v = \frac{A_a + A_v}{A_{PI} + \overline{A_v}} \qquad (2)$$

This simple algebraic relationship shows that the $R_v$ is smaller than $R_0$. Since these ratios largely determines historical cloud albedo change, a lower ratio suggests a weaker aerosol-cloud interaction and reduced cloud cooling in historical transient simulations (Schmidt et al., 2012).

Furthermore, volcanic sulfate aerosols fluctuate over time, their effects on aerosol-cloud interaction and cloud albedo also varies as a function of time. During periods of low volcanic activity, volcanic sulfate aerosols remain below the historical average ($\overline{A_v}$), leading to a reduced $R_v$ and a relative dimmed cloud cooling. Conversely, active volcanic periods result in increased volcanic $SO_2$ emissions (strengthens $A_v$), resulting in larger $R_v$, and amplifying aerosol-cloud interactions. These variations highlight the necessity of accurately representing volcanic aerosols in climate simulations."

Line 81: "version 3" is stated here but everywhere else in the manuscript discusses "version 2". Please reconcile.

We thank reviewer pointed it out and corrected this expression.

Fig. 1: The SO2 cloud in the stratosphere is just too unrealistic and evokes a polar stratospheric cloud rather than a gas. Perhaps this would be better depicted as just the word 'SO2' with no cloud, as with H2SO4.

We thank reviewer pointed it out and revised Figure 1.

Lines 100-116: I feel this paragraph is separate from "volcanic forcing representation" and may be better spun off into a separate sub-section on E3SMv2 itself as compared to volcanic experiments. Also, because aerosol-cloud interactions are imperative for this study, it would be beneficial to add 1-2 lines on E3SM's CCN activation.

We appreciate reviewer's suggestions. A new subsection (2.1) has been added to introduce the E3SM and its Aerosol and Cloud Parameterizations. The volcanic forcing treatment has been moved to subsection 2.2.

Lines 118-135: This paragraph feels longer and more technical than it needs to. I would cite the dataset itself if possible (or a paper that describes it) and give a simpler description, rather than mentioning many instruments it incorporates. It would seem to be the pre-SAGE period that overwhelmingly matters most here. Is this using the standard CMIP6 volcanic emissions dataset? If so, there's no proper paper to cite, but I suggest one way to cite the data webpage below. But I'm not familiar with the AER-2-D use here, so wonder if this is E3SM's own method instead. If this is not a standard CMIP6 volcanic aerosol prescription, this should be stated.
IACETH, 2017: CMIP6 Stratospheric Aerosol Dataset (SAD) v3. Institute for Atmosphere and Climate, ETH Zurich, Earth System Grid Federation, accessed 3 August 2024, https://doi.org/10.22033/ESGF/input4MIPs.1681.

We appreciate the reviewer's suggestion and have added the appropriate citation for the dataset. However, we believe it is important to retain the description of how the stratospheric aerosol optical depth (SAD) data was produced. The pre-SAGE period has substantial uncertainty, and the use of the AER-2D model in reconstructing SAD data contributes to this uncertainty. Given its relevance to our study, we prefer to keep this description to provide necessary context on potential sources of variability in the stratospheric forcing data. Additionally, providing a detailed explanation of the model and methodology aligns with GMD's journal scope.

Line 138: Are there really only two volcanic events? In Fig 2 there are two unexplained peaks between the two mentioned eruptions in the V2-CMIP6 simulations that presumably are using this data method.

We agree with the reviewer. While Arfeuille et al. (2014) only document two volcanic events during this period, there are two unexplained moderate peaks between the Agung (1963) and Fuego (1974) eruptions in the V2-CMIP6 simulations. Despite conducting a literature search, we could not find conclusive evidence attributing these peaks to specific volcanic eruptions in CMIP6 SAD dataset. This uncertainty highlights the need for a detailed description of the dataset in Section 2.2 to ensure consistency in the stratospheric aerosol forcing data. Additionally, one of the objectives of this study is to conduct a thorough evaluation of the CMIP6 stratospheric SAD dataset for the pre-satellite era, addressing potential discrepancies and uncertainties in volcanic aerosol records.

Line 137-144: Somewhere here, as well as elsewhere in the study, should explain that – if I understand right – the focus is particularly on explosive eruptions that emit (primarily?) into the stratosphere. Effusive eruptions more directly bring sulfur into the troposphere but do not seem to be included in either of datasets for prescribed aerosol forcing or prescribed SO2, so this should be clarified.

In this study, we focus on explosive volcanic eruptions. Sorry for the confusion.

Lines 161-2: There have been efforts to correct volcanic aerosol impacts in E3SMv2/MAM4, e.g. Brown et al., 2024. Maybe this particular version wasn't used here, but hopefully there don't wind up with a bunch of separate E3SM versions for representing the same processes.
Brown, H. Y., Wagman, B., Bull, D., Peterson, K., Hillman, B., Liu, X., ... & Lin, L. (2024). Validating a microphysical prognostic stratospheric aerosol implementation in E3SMv2 using observations after the Mount Pinatubo eruption. Geoscientific Model Development, 17(13), 5087-5121.

We appreciate the reviewer's comment. In Brown et al. (2024), the authors modified the coarse-mode aerosol standard deviation and size range to reduce gravitational settling of aerosol particles, improving the simulation of sulfate aerosol burden in the stratosphere following the 1991 Mt. Pinatubo eruption. A similar approach was previously used in Mills et al. (2016). However, such modifications can potentially introduce unintended effects, including unrealistic dust and sea salt distributions, which may impact broader climate (Visioni et al. 2018). Since this study aims to improve volcanic forcing representation without significantly altering tropospheric climate processes, we do not incorporate the MAM4 modifications used in Brown et al. (2024). Instead, we focus solely on incorporating volcanic SO2 emissions to enhance the model's volcanic aerosol treatment. Furthermore,

it is important to use unchanged MAM4 and E3SMv2 configurations to provide an apple-to-apple comparison to evaluate the impacts of the change of volcanic sulfate aerosol representation on simulated aerosol direct and indirect effects during middle of 20th century.

Lines 180-1: For the SAGE period the original methods feeds into the model observed aerosol extinctions, so it would be highly possible this performs more accurately than the new SO2-emission method, at least if stratospheric aerosol falling into the troposphere does not cause a prominent forcing. But for the mid-century period both methods rely on (presumably quite poorly constrained) SO2 estimates. For this period we expect the new version is improved specifically because of 1) the added explosive SO2 injection events and 2) the addition of aerosol-cloud interactions caused by CCN resulting from explosive eruptions. ARI after Fuego and Agung will be different but for this this isn't a reason to expect improvement. Could the authors please state if my understanding here is correct? Potentially the text should clarify some of this.

We appreciate the reviewer's insights and confirm that their understanding is largely correct. During the SAGE period, the original method incorporates observed aerosol extinctions, which likely provides a more accurate representation of stratospheric aerosol-radiation interactions compared to the SO2-emission-based approach—assuming that stratospheric aerosol sedimentation into the troposphere does not significantly affect radiative forcing. However, for the mid-century period, both methods rely on SO2 emission estimates, which are inherently uncertain. The key improvements in our approach stem from (1) the inclusion of updated explosive SO2 injection events, ensuring a more physically consistent representation of volcanic forcing, and (2) the incorporation of aerosol-cloud interactions through the cloud condensation nuclei (CCN) effect from volcanic sulfate aerosols, which was absent in the original approach. We have revised the text to clarify these aspects as suggested.

Tables 1 and 2: All these eruptions appear to be explosive rather than effusive. Can the authors please make it clear what the focus is in the captions?

We thank reviewer's suggestion. All eruptions we discussed during 1940-1980 are explosive eruptions. We corrected them in the captions.

Table 1 specifically: Since this says "(CMIP6)", can the authors please confirm this is the method used in the CMIP6 protocol and not just the CMIP6 E3SM simulations, and if not, specify?

We appreciate reviewer's concern and we can confirm this is the method used in the CMIP6 protocol (Rieger et al., 2020; Thomason et al., 2018).

Table 2 specifically: Is it confidently known that these eruptions emitted into the stratosphere, or are these very crude estimates of what occurred? The stratospheric

fraction is expected to be the most important, as it is the longest lived and most globally distributed – is there no stratospheric amount available?

We appreciate the reviewer's question and recognize the need for further clarification in the manuscript. Table 2 lists the recorded explosive eruptions from the volcanEESM dataset for the period 1940–1979. This dataset, developed through the NCAR/UCAR Atmospheric Chemistry and Modeling Visiting Scientist Program and the University of Leeds School of Earth and Environment, provides detailed information on historical volcanic eruptions, including eruption dates, locations, injection height ranges, and SO2 emission amounts. However, whether the injected volcanic sulfate aerosols reach the stratosphere or remain in the troposphere is ultimately determined by the E3SM simulation, which depends on the modeled tropopause height and atmospheric circulation patterns.

In Section 3.1, we compare the simulated Agung eruption with results from other studies, demonstrating that the new approach better captures volcanic sulfate aerosol-radiation interactions (ARI) in terms of both magnitude and spatial distribution than the CMIP6 prescribed stratospheric forcing. This comparison supports the validity of the emission-based approach in improving the representation of volcanic aerosol forcing.

Lines 250-90: Most of the material here is redundant with material in the Introduction. For instance VARI, VACI, "volcanic quiescent warming effect", and "volcanic surplus cooling effect" are all being introduced as if for the first time, but this is not the case. I'd recommend omitting this section and moving unique information into the Introduction's version, as this isn't really methodology.
Lines 270-277: I find this confusing, as most experiments and nearly all results are about volcanic aerosol impacts on historical climate but this largely is about anthropogenic aerosols and PI. I'm not convinced it's worth focusing on interactions between volcanic and anthropogenic aerosols, and there aren't experiments here to isolate these from volcanic aerosol impacts on their own, which would entail also varying anthropogenic aerosol emissions. I feel the manuscript would be stronger if it kept a clearer focus on how explosive volcanic aerosols affect mid-20th century temperature development. The PI background state is worth testing, but I feel reporting this can be more briefly folded into the results only.
Figure 3: As above, I find this figure distracting from the main experiments and unnecessary. I also don't understand what it depicts. Adding background volcanic aerosols would, if interactive, mean more CCN and more cloud scattering, not less. The PI background state is certainly worth testing but is not a focus in any other figure. If this authors want to keep the figure, I hope they can make it and its importance clearer. The capitalization should also be consistent.

We appreciate the reviewer's suggestion and have revised the manuscript accordingly. We have removed this section and Figure 3 to maintain a clearer focus on the role of explosive volcanic aerosols in mid-20th century temperature development. The discussion of pre-industrial (PI) background volcanic aerosols has been incorporated into the **Introduction**.

These changes improve the manuscript's clarity and align better with the study's primary objectives.

Table 3: Since "V2-IVA-NPI" isn't used in any Figures, I'm not convinced it's worth adding here or describing in the methods, as the paper would be easier to digest this way. I do, however, think this would be suitable for a paragraph in Results, e.g. "We repeated the V2-IVA experiment but branched off from a preindustrial control simulation having no volcanic emissions. This altered the results such that [...]", etc.

We appreciate the reviewer's suggestion and have revised the manuscript accordingly.

Lines 340-60: These first two paragraphs seem largely more suited to Methods, as they're mostly just saying what's in the volcanEESM datasets. Contrarily, the Methods has Figure 2 and lines describing it (183-9) that are instead results in that they show sulfate properties calculated by the model rather than SO2 inputs.

We appreciate the reviewer's comments. The figure and the two paragraphs describe the temporal evolution of sulfate aerosol distribution during the 1940–1980 period. In addition to providing information on sulfate aerosol presence following major volcanic eruptions, they also serve as evidence that volcanic aerosols descend into the troposphere after lingering in the stratosphere. Given their role in demonstrating these key aerosol transport processes, we believe they are best suited for Section 3.1.

Lines 401-26: I recommend splitting off the radiative forcing results into its own subsection, as this section is very long right now.

We thank reviewer's suggestion and split this part as Section 3.2.

Figure 7: I feel strongly that the net forcing should be shown rather than its shortwave and longwave components, as it's otherwise very hard to gauge whether we should expect these forcings to cause different surface temperature developments. The SW and LW would be appropriate for the supplement if the authors want to cite them as showing clearer differences than the net effect, but do not seem strictly necessary. Second, because these results are critical for the next section and the paper's conclusions, there should be uncertainty bars. Third, because the paper focused on aerosol-cloud interactions it's important to show the net cloud radiative effects (all sky – clear-sky), as currently it's impossible to much this varies among experiments. And fourth, the LW, if shown, should be positive, as it is offsetting rather than augmenting the SW forcing.

We appreciate the reviewer's detailed feedback on Figure 7. In response, we have revised Figure 7 (now Figure 6, following the removal of Figure 3). The updated figure now includes clear-sky shortwave forcing at the top of the atmosphere (TOA), net radiative forcing at TOA, and net cloud forcing at TOA.

The clear-sky shortwave forcing is included to illustrate aerosol-radiation interactions, providing further validation of our emission-based volcanic eruption representation. The net radiative forcing at TOA assesses the overall impact of volcanic eruption representations on the climate energy balance. The net cloud forcing at TOA examines how cloud radiative effects respond to volcanic sulfate aerosol changes.

As requested, we now show the ensemble mean of forcings as solid lines, with the one-standard-deviation range represented as color shading to indicate uncertainty. The figure description and context has been revised accordingly.

Line 492: Does "V2" mean "V2-CMIP6"?
Yes. We corrected that.

Lines 501-10: First, it can't seriously be ignored that there's a huge remaining bias in surface temperature development. Following Agung the revised model experiment nevertheless descends into a heavy bias just like the original experiment. Given the stated goal was to fix this bias, the result is largely negative and this really must be fairly explained. Second, shouldn't the surface temperature differences relate to the volcanic quiescent warming and active cooling processes the authors have been discussing? Perhaps the authors can put these results into that context? Third, it doesn't seem like Table S1 needs to exist, as most of the values in it are already stated in this paragraph. Maybe just bring in the rest?
Line 502: Is the V2-IVA line slope here statistically significantly distinct from zero? Should it be?
Line 506: "V2-IVA" mitigates this temperature drop"? The temperature drop is nearly exactly the same.
Line 507: Which observations are used in the comparison? Fig 10 shows 3 different datasets. Is this one or particular or an average?

We appreciate the reviewer's insightful comments regarding the remaining bias in simulated surface temperature anomalies. We have revised the discussion of Figure 10 to explicitly acknowledge that, while V2-IVA improves the interannual variability of surface temperature by incorporating additional volcanic eruptions (Helka in 1947 and Bezymianny in 1956), it does not significantly reduce the overall cold bias present in V2-CMIP6. We now highlight that although V2-IVA produces slightly warmer temperatures after the Mt. Agung eruption, it fails to capture the observed temperature rebound in the early 1970s, and both simulations still exhibit a cooling trend over the two-decade period.
We removed Table S1.

"Figure 10 presents the simulated surface temperature anomalies compared to observations, with three observational datasets shown as solid gray to black lines. We select the HadCRUT5-Analysis product (Morice et al., 2021), NOAA's National Climatic Data Center (NCDC) NCEP reanalysis dateset, and NASA Goddard Institute Surface Temperature (GISTEMP) dateset (Hansen et al. 2010). Between 1940 and 1959, observations exhibit interannual variability of up to 0.25°C, along with a moderate cooling trend. In contrast, V2-CMIP6 produces a nearly flat temperature curve during this period. V2-IVA improves the interannual variability by

incorporating two additional volcanic eruptions, Helka (1947) and Bezymianny (1956), which introduce episodic cooling events. As a result, the correlation coefficient between the simulated and observed temperature anomalies increases from 0.15 in V2-CMIP6 to 0.38 in V2-IVA, suggesting an improved representation of temperature fluctuations. However, the mean surface temperature anomaly simulated by V2-IVA is only slightly warmer (by 0.02°C) than that of V2-CMIP6, indicating that the main cold bias remains largely unchanged.

Between 1960 and 1979, the observations show a moderate temperature drop (up to 0.3°C) following the Mt. Agung eruption, but temperatures quickly recover to pre-eruption levels in the early 1970s, with an overall weak warming trend. In contrast, while V2-IVA simulates a slightly warmer temperature anomaly after Mt. Agung compared to V2-CMIP6, the model does not capture the observed temperature rebound. Instead, both simulations exhibit a continued cooling trend over this two-decade period. This result suggests that while the revised volcanic forcing in V2-IVA slightly moderates the excessive cooling seen in V2-CMIP6, it does not clearly correct the underlying bias, indicating that additional factors beyond volcanic forcing may contribute to the mid-20th-century temperature discrepancies.
"

Fig. 10: There doesn't seem to be any explanation (or citation) of the observations in the text. Also, the colors in the legend don't seem to match – possibly NOAA NCDC is one of the grey lines, despite being blue in the legend?

We thank reviewer to point out this legend mistake. We corrected the errors and added the citations about the observational datasets.

Lines 546-602: I feel this section should precede the section on historical temperature, as the cloud forcing is a contributor to temperatures and the temperature anomalies are really the culminating results of this study. I also feel this section would benefit from a figure on cloud changes. Perhaps CCN density and/or net CRE could be plotted as a function of latitude only? Okay if not, as long as the net CRE is worked into Fig. 7 and the reader can somehow appreciate whether CCN are different dissimilar between experiments or not. One potential issue worth checking is that if the explosive eruption impact on CCN is substantial but only in high clouds, the targeted clouds could have balanced shortwave and longwave effects (unlike low clouds which predominantly cause cooling via SW, but already have many CCN).

We appreciate reviewer's suggestion. Instead of CCN, we use CDNC to represent cloud droplet changes. The zonal mean of anomaly difference in CDNC, cloud cover, and cloud liquid water path have been shown in new Figure 7. As reviewer suggested, we move this subsection right after the simulated forcing subsection to discuss the link between the simulated cloud forcing change with simulated cloud property changes.

"This section examines cloud property changes over time to support the cloud forcing differences discussed in Section 3.2. The difference in net cloud forcing anomalies between the V2-IVA and V2-CMIP6 ensembles is 0.11 W/m$^2$ for the 1940–1959 period and -0.07 W/m$^2$ for the 1960–1979 period. These changes align with variations in volcanic activity levels during these periods, with relatively low volcanic emissions in 1940–1959 and elevated emissions in 1960–1979. The shortwave cloud forcing exhibits a similar pattern (Table 4).

In the V2-IVA experiment, volcanic sulfate aerosols ($A_v$) actively participate in aerosol-cloud interactions, modifying cloud properties in response to fluctuations in volcanic emissions. During 1940–1959, the volcanEESM inventory recorded two eruptions, Helka (1947) and Bezymianny (1956), contributing a total of 6.2 Tg $SO_2$ emissions, or an average of 3.1 Tg per decade. This is lower than both the historical average (7.0 Tg per decade) and the 1850–1899 climatology (6.5 Tg per decade), indicating a reduction in volcanic sulfate emissions during this period. Since the total aerosol burden in the atmosphere is determined by both historical aerosols increase ($A_a$) and volcanic aerosols ($A_v$), a lower $A_v$ value in this period partially compensates the $A_a$ increase and lowers aerosol increase ratio $R_v$ (equation 2). This lower $R_v$ results in relative lower cloud droplet number concentration (CDNC), cloud cover, and cloud liquid water content leading to weaker aerosol-cloud interactions in the historical transient simulation (Figure 7, left column). These reductions occur relatively uniformly across latitudes rather than being concentrated in specific regions.

In contrast, the 1960–1979 period experienced significantly higher volcanic $SO_2$ emissions, totaling 16.4 Tg, or 8.2 Tg per decade (Table 2), which exceeds the 1850–1899 climatology of 6.5 Tg per decade. The larger $A_v$ value during this period lowers the $R_v$ ratio, meaning that volcanic aerosols contribute more substantially to the total aerosol burden, as well as other aerosol increase. The increase in volcanic sulfate aerosols results in a 28% increase in low cloud fraction anomaly, a 5% increase in cloud liquid water path anomaly, and a 1% increase in vertically integrated CDNC in V2-IVA compared to V2-CMIP6 (Figure 7. right column and Table 4). These cloud property changes support a cooling effect on net cloud forcing of -0.08 $W/m^2$. The simulated cloud fraction anomaly values are reasonable and in line with the study that evaluates CMIP6 cloud fraction variations across different climate models (Vignesh et al., 2020).
„

[Figure]

Figure 7. Simulated difference in cloud property anomalies between V2-IVA and V2-CMIP6 ensembles for 1940-1959 period (left column) and 1960-1979 (right column). The comparison includes (top row) zonal mean vertical integrated cloud droplet number concentration (CDNC) in units of #/m², (middle row) low cloud

cover, and (bottom row) cloud liquid water path (LWP) in units of kg/m2. The y-axis in each panel represents latitude (degree).The red dotted points represent the difference are significant at 95% confidence level.

Line 602 and also Lines 694-5: "Low bias spreading" and "low bias spread"? The temperatures are too cool (a low bias) but I don't understand why these instances refer to a 'spread'.
We appreciate reviewer's suggestion. As new volcanic forcing representation doesn't clearly improved the temperature low bias, we deleted this sentence in the manuscript.

Table 4: Can the authors please show some info on CCN number density? As is I don't see clear evidence of aerosol-cloud interactions being directly affected by falling sulfate, which seems to be one of the main motivations for this study. Cloud properties will be radiatively affected by the aerosols in the stratosphere even with no CCN change. Even better would be a figure on this. Also, there are two different reds in this table.

We appreciate reviewer's suggestion. In the revised manuscript, we demonstrated the change of cloud drop number concentration, cloud cover and liquid water content.

Lines 626-652: I don't feel this is worth its own section, given it's quite independent from the main results on 20th century impacts. I think it would make more sense to add the most pertinent results to the preceding cloud forcing section. I also think it makes the most sense to show these results as in Table 4, but comparing V2-IVA-NPI (which I don't think needs its own name and mentions elsewhere) to V2-CMIP6 (rather than V2-IVA-NPA vs V2-IVA). Then Table 4 could simply show both comparisons and the whole supplement can be omitted if the other reviewer(s) don't request new plots there.

We thank reviewer's suggestion and remove this part to supplement.

Lines 656-709: The Discussion section largely repeats the Results. There should be some discussion of caveats and also reasons why the model improvement is not larger (see major comments). Clearly the improvement is not as large as would have been hoped, as the surface temperature bias grows after Agung nearly as much as previously. I'd really like the authors to discuss this and why the effect is not larger. E.g. Are there simply far too many CCN for slowly falling stratospheric aerosols to matter? Are the cloud forcing changes too balanced between SW and LW? Are the volcanic aerosol being drawn down over narrow polar regions rather than broad cloud-rich areas?

We appreciate reviewer's suggestion and rewrite the whole section

"This study evaluates the impact of an improved volcanic sulfate aerosol representation in E3SM version 2 on mid-20th-century climate simulations. By implementing an emission-based approach that accounts for both volcanic sulfate aerosol-radiation interactions (VARIs) and volcanic sulfate aerosol-cloud interactions (VACIs), we aimed to address the substantial surface

temperature cold bias observed in E3SM historical simulations. The revised model (V2-IVA) exhibits moderate improvements in surface temperature variability and a slightly warmer simulated climate compared to the default CMIP6 treatment (V2-CMIP6). The inclusion of additional volcanic events, such as Helka (1947) and Bezymianny (1956), enhances the representation of temperature fluctuations, while the revised Mt. Agung eruption (1963) intensity and the more realistic treatment of VACIs contribute to the slightly warmer climate simulated in V2-IVA. However, despite these improvements, the cold bias in historical simulations remains largely uncorrected.

One key finding is that cloud forcing anomalies in V2-IVA are more consistent with expected aerosol-cloud interaction processes. During the 1940–1959 period, when volcanic sulfate emissions (Av) were below the historical average, cloud forcing anomalies were reduced relative to V2-CMIP6, leading to a weaker cloud cooling effect (0.11 W/m$^2$). Conversely, during the 1960–1979 period, elevated volcanic sulfate emissions strengthened aerosol-cloud interactions, resulting in a stronger cooling effect (-0.07 W/m$^2$). This aligns with theoretical expectations based on the aerosol increase ratio ($R_v$), as described in Equation 2. While these cloud forcing changes help explain some differences in the temperature response, they are not sufficient to fully resolve the cold bias.

Several factors may explain why the temperature cold bias remains substantial. First, the overall aerosol cooling effect in E3SM is approximately twice as strong as the CMIP6 multi-model mean (Golaz et al., 2022). This discrepancy may be attributed to the absence of updates in aerosol wet removal processes (Shan et al., 2021, 2024), which could lead to an overestimation of aerosol burden and associated cooling effects. Additionally, while E3SMv2 implements the O3v2 model to better represent stratospheric ozone (Tang et al., 2021), its treatment of tropospheric ozone lacks historical variability, potentially missing a warming contribution from tropospheric ozone changes (Tang et al. in preparation). Another limitation is the use of the Liu and Penner (2005) scheme for homogeneous and heterogeneous ice nucleation in cirrus clouds, which considers only sulfate aerosol particles in the Aitken mode. In reality, volcanic aerosols in the stratosphere rapidly grow from Aitken to accumulation and even coarse mode when large eruptions occur, suggesting that ice nucleation from these larger particles should be considered, as implemented in CESM2 (Visioni, 2017). Furthermore, one of the major sources of background stratospheric sulfate, carbonyl sulfide (OCS), is not included in the current study. Missing this source may lead to an underestimation of stratospheric sulfate aerosol levels, particularly during periods of low volcanic activity, potentially accounting for as much as one-third of the background sulfate burden in the stratosphere (Mills et al., 2016). Addressing these limitations in future model versions could further improve the representation of aerosol-climate interactions and help reduce the persistent temperature bias"

---

## Author Comment (AC2)

General comments:

The paper is devoted to reducing the bias in climate simulations by improving the representation of volcanic aerosols in CMIP6 models. This objective failed, as no notable improvement was obtained. However, the research has a promise. The authors implemented a sulfur cycle and the formation of volcanic aerosols. This is advantageous compared to the prescription of monthly aerosol fields, as in most CMIP6 simulations. This approach is not new and has been widely implemented in different studies in the last 20 years. The second mechanism employed to reduce the bias is the indirect effect of volcanic aerosols on tropospheric clouds. This is good but not a new idea. Ulrike Lohman worked on detecting this effect after major volcanic eruptions. In addition, this effect is poorly described by the models, so the first thing the authors have to show is that their model can reproduce this effect. It, by itself, is a complex task. The authors claimed that volcanic aerosols affected low-level clouds and said nothing about the impact on upper-level clouds. This is hardly believable. Volcanic aerosols will first affect cirruses and upper-tropospheric clouds. I doubt any volcanic aerosols could reach the lower troposphere, and, in any case,  their contribution will be negligible in comparison with tropospheric (natural and anthropogenic) aerosols. The figures are reasonably well prepared, although showing the globally averaged tropopause height is useless. The authors should more clearly define the statistical significance of the results and make ensemble calculations to reach statistical significance. The text is poorly written, with repetitions, incorrect terminology, and poor English.

All our responses are in blue color

We sincerely appreciate the reviewer's thoughtful and detailed feedback, which has helped us improve the clarity and scientific focus of our study. We acknowledge the concerns regarding the model's improvements and methodological choices, and we have carefully revised the manuscript to better communicate the study's objectives and findings. Below, we provide a point-by-point response to the reviewer's comments.

**Model Improvement & Scientific Contribution**
We recognize that our implementation did not fully resolve the mid-20th-century surface temperature cold bias. In response to this concern, we have substantially revised the manuscript to better clarify the purpose of this study. The primary goal of our work is to assess the impact of an improved volcanic sulfate aerosol representation on climate simulations, particularly in its influence on aerosol-cloud interactions. We have updated the title to **"Assessing the Climate Impact of an Improved Volcanic Sulfate Aerosol Representation in E3SM"** to better reflect this objective.

We also acknowledge that our initial presentation may have overemphasized the extent of improvement achieved. To address this, we have refined the most parts of the manuscript to provide a more clear and reliable assessment of our results. Rather than focusing solely on reducing the cold bias, we now highlight the more realistic volcanic sulfate aerosol representation and evaluate the implementation's impact on historical cloud forcing changes. We also discuss the possible next steps to further improve E3SM's bias. The abstract has been revised accordingly to reflect these clarifications.

"Accurately simulating historical surface temperature variations is essential for evaluating climate models, yet many struggle to reproduce the mid-20th-century temperature trends associated with significant volcanic eruptions. This study examines the impact of volcanic sulfate aerosol representation on these biases using the Energy Exascale Earth System Model (E3SM). The standard CMIP6 protocol prescribes volcanic forcing through radiative perturbations, omitting volcanic aerosol-cloud interactions (VACIs). Here, we implement an emission-based approach with an updated volcanic eruption inventory that directly incorporates volcanic sulfur dioxide ($SO_2$) emissions, enabling a more process-based representation of volcanic forcing. This approach leads to improved surface temperature variability and a modest reduction in cold biases between 1940 and 1980 compared to the CMIP6 setup. Additionally, we assess cloud property responses to a more realistic volcanic sulfate aerosol representation, which weakens cloud-induced cooling during periods of lower volcanic activity. However, despite these refinements, a significant temperature cold bias remains, indicating that further improvements in aerosol microphysics, cloud processes, and model parameterizations are needed to fully resolve this issue in E3SM."

we hope these revisions address reviewer's concerns.

**Novelty of the Approach**

We appreciate the reviewer's point that an emission-based approach to volcanic sulfate aerosol representation is not entirely new, as prior studies, including those by Lohmann et al. (2002), have explored aerosol-cloud interactions from volcanic eruptions. However, our study specifically integrates this approach within E3SM and evaluates its long-term impact on mid-20th-century historical transient climate simulations. Given the significant temperature biases in CMIP6-era models, we believe this assessment provides valuable insights.

Additionally, we build upon the work of Schmidt et al. (2012), who emphasized that background volcanic degassing has long-term climate implications. Our study extends this discussion by assessing the role of explosive volcanic eruptions in shaping aerosol-cloud interactions. Since the standard CMIP6 protocol prescribes volcanic forcing primarily through radiative perturbations, neglecting aerosol-cloud interactions, our approach offers a novel contribution by explicitly quantifying these effects within a fully coupled Earth system model. We have revised the introduction to better situate our work in the context of prior research and clarify its contributions.

**Volcanic Aerosol Effects on Upper vs. Lower Troposphere Clouds**

We appreciate the reviewer's suggestion to consider the impact of volcanic aerosols on high clouds and have now included an additional analysis in **subsection 3.3 and Table 4**. While our study primarily focuses on interactions between volcanic sulfate aerosols and lower-tropospheric clouds, we agree that a comprehensive evaluation should also assess upper-level cloud impacts.

Lohmann et al. (2002) conducted foundational work on the influence of the Pinatubo eruption on homogeneous ice nucleation and cirrus clouds. However, observational studies such as Luo et al. (2002) found no significant climate feedback from aerosol–cirrus–radiative interactions when

examining multiple satellite products. This discrepancy highlights the complexity of these processes and the challenges in representing them in models.

From a modeling perspective, the response of cirrus clouds to volcanic sulfate aerosols is strongly dependent on the choice of ice nucleation parameterization. The current E3SM MAM4 aerosol scheme (Liu et al., 2016) and the ice nucleation scheme based on Liu and Penner (2005) may not be well-suited for investigating these effects in detail. We acknowledge this limitation and have added a discussion of it in the revised manuscript.

Furthermore, Figure 3 now includes the vertical distribution of sulfate aerosols, which clearly shows that a substantial fraction of volcanic sulfate can descend into the middle and lower troposphere, where it influences cloud microphysical processes. We have also clarified this in the revised discussion.

**Statistical Significance & Ensemble Considerations**

We appreciate the reviewer's comments regarding the need for clearer statistical significance testing. To address this, we have revised our figures and tables as follows:

- **Figure 6**: Now includes uncertainty markers to indicate ensemble uncertainty range.
- **Figure 7 and Table 4**: Now include statistical significance indicators to clarify where differences between simulations are meaningful.

We believe these additions improve the robustness of our analysis.

**Language & Clarity**

We acknowledge the reviewer's concerns regarding clarity and readability. The manuscript has undergone extensive revisions to reduce redundancy, refine terminology, and improve overall readability.

Specific comments:

L105: macrophysics > microphysics

We thank reviewer's suggestion and correct it.

L148: processes > microphysics

We thank reviewer's suggestion and rewrite this part.

L262: Not only LW

We thank reviewer's suggestion and rewrite this part.

L279-285: Improve language

This part of manuscript has been rewrite.

L287-290: Volcanic aerosols penetrate into the troposphere mostly through tropopause folds and in high-latitudes

We thank reviewer's suggestion.

L362: optical properties of the atmosphere in the stratosphere

We thank reviewer's suggestion

L388: light extinction > aerosols extinction

We very appreciate this detailed correction.

Helka > Hekla in multiple places

We very appreciate this detailed correction

L393: response > extinctionon

We thank reviewer's suggestion

L408: mentioned > showed

We thank reviewer's suggestion

L412: Panel c > Figure 7c

We very appreciate this detailed correction

L417: panel d > Figure 7d

We very appreciate this detailed correction

L422-426: Clarify the text

This part has been rewrite.

L488: dimmer volcanic eruptions?

We changed the word choice here.

L491 anomalies are warmer > anomalies are greater

We changed the word choice here